# A miR-150/TET3 pathway regulates the generation of mouse and human non-classical monocyte subset

Dorothée Selimoglu-Buet [1], Julie Rivière[1,2], Hussein Ghamlouch[1,2], Laura Bencheikh[1,2], Catherine Lacout[1,2], Margot Morabito[1,2], M'boyba Diop [3], Guillaume Meurice[3], Marie Breckler[3], Aurélie Chauveau[1,4], Camille Debord[1,5], Franck Debeurme[1,2], Raphael Itzykson[1,6], Nicolas Chapuis[7], Christophe Willekens [1,8,9], Orianne Wagner-Ballon[10], Olivier A. Bernard[1,2], Nathalie Droin [1,2,3] & Eric Solary[1,2,9]

Non-classical monocyte subsets may derive from classical monocyte differentiation and the proportion of each subset is tightly controlled. Deregulation of this repartition is observed in diverse human diseases, including chronic myelomonocytic leukemia (CMML) in which non-classical monocyte numbers are significantly decreased relative to healthy controls. Here, we identify a down-regulation of hsa-miR-150 through methylation of a lineage-specific promoter in CMML monocytes. Mir150 knock-out mice demonstrate a cell-autonomous defect in non-classical monocytes. Our pulldown experiments point to Ten-Eleven-Translocation-3 (TET3) mRNA as a hsa-miR-150 target in classical human monocytes. We show that Tet3 knockout mice generate an increased number of non-classical monocytes. Our results identify the miR-150/TET3 axis as being involved in the generation of non-classical monocytes.

[1] INSERM U1170, Gustave Roussy Cancer Center, 94805 Villejuif, France. [2] Université Paris-Sud, Faculté de Médecine, 94270 Le Kremlin-Bicêtre, France. [3] INSERM US23, CNRS UMS 3655, Gustave Roussy Cancer Center, 94805 Villejuif, France. [4] Laboratoire d'Hématologie, Centre Hospitalier Régional Universitaire, 29200 Brest, France. [5] Laboratoire d'Hématologie, Centre Hospitalier Régional Universitaire, 44000 Nantes, France. [6] Département d'Hématologie, Hôpital Saint-Louis, Assistance Publique-Hôpitaux de Paris, 75010 Paris, France. [7] Département d'Immuno-Hématologie, Institut Cochin, 75014 Paris, France. [8] Université Paris Diderot, 75004 Paris, France. [9] Département d'Hématologie, Gustave Roussy Cancer Center, 94805 Villejuif, France. [10] Département d'Hématologie et d'Immunologie Biologiques, Hôpital Henri-Mondor, 94010 Créteil, France. These authors contributed equally: Nathalie Droin, Eric Solary. Correspondence and requests for materials should be addressed to D.S.-B. (email: dorothee.selimoglubuet@gustaveroussy.fr) or to E.S. (email: eric.solary@gustaveroussy.fr)

Peripheral blood monocytes are components of the mono-nuclear phagocyte system that is involved in rapid recognition and clearance of invading pathogens and in tissue integrity maintenance. In mice and humans, the peripheral blood monocyte pool is phenotypically and functionally heterogeneous. Using Ly6C surface antigen, two main monocyte subsets are depicted in mice[1]. Ly6C$^{high}$, CX3CR1$^{low}$, CCR2$^{high}$, CD43$^{low}$, CD62L$^+$ classical monocytes are progenitors of inflammatory and some tissue-resident macrophages. Ly6C$^{low}$, CX3CR1$^{high}$, CCR2$^{low}$, CD43$^{high}$, CD62L nonclassical monocytes patrol the resting endothelium to maintain vascular homeostasis[2], protect against the seeding of metastatic tumor cells, and populate tissues to contribute to wound healing[3]. Based on cell surface expression of CD14 and CD16, human circulating monocytes are divided into three subsets[4]. Classical CD14$^+$, CD16$^-$ monocytes, which represent roughly 85% of total human monocytes, are analogous to Ly6C$^{high}$ mouse monocytes, express high level of the chemokine receptor CCR2, show a strong inflammatory response to lipopolysaccharides, and are rapidly recruited to inflamed and infected tissues. They are distinguished from CD14$^+$, CD16$^+$ intermediate monocytes and CD14$^{low}$, CD16$^+$ nonclassical monocytes, which are more probably analogous to Ly6C$^{low}$ monocytes, express higher levels of CX3CR1 receptor, and remain longer in the blood to patrol on the luminal part of the endothelium wall[2,5].

Monocyte subsets arise from a multipotent hematopoietic stem cell via a series of progenitors with increasingly restricted lineage potential[6]. Detailed steps of this differentiation pathway remain unclear as recent studies have challenged traditional hierarchical models by suggesting that some progenitors could make lineage commitment earlier than initially thought[7–9]. Accordingly, two independent pathways of monocyte generation were recently depicted in mice, leading to functionally distinct Ly6C$^{high}$ monocytes whose production varies in steady state and stress condition[10,11]. The origin of circulating monocyte subsets has long been controversial until a consensus model emerged supporting a hierarchical relationship in which most if not all Ly6C$^{low}$ monocytes derive from Ly6C$^{high}$ monocytes and the transcription factor Nr4a1 is the master regulator of the differentiation and survival of Ly6C$^{low}$ monocyte subset[12,13]. Gene expression patterns depicted in human monocyte subsets also suggest a common origin[14,15]. Finally, in vivo deuterium labeling supports a model of sequential transition from classical to intermediate and nonclassical monocytes, at steady state and in response to endotoxemia[16]. However, the molecular mechanisms that drive the generation of nonclassical monocytes in humans are only partly understood.

Some pathological conditions offer opportunities to explore human monocyte biology and more specifically the relationships between human monocyte subsets. A depletion of the slan-positive nonclassical monocytes is observed in patients with hereditary diffuse leukoencephalopathy with axonal spheroids, a rare neurologic disease associated with inactivating mutations in the colony-stimulating factor-1 receptor (CSF1R) tyrosine kinase domain[17]. Conversely, infectious and inflammatory diseases promote an increase in the fraction of intermediate and non-classical monocytes[17–19]. We detected an abnormal repartition of monocyte subsets in the blood of patients with a chronic myeloid malignancy, especially those with a chronic myelomonocytic leukemia (CMML)[20]. CMML is a clonal hematopoietic stem cell malignancy related to the accumulation of recurrent somatic mutations in epigenetic regulators, splicing genes, and signaling genes, mostly of the RAS pathway[21–24]. Flow cytometry analysis of peripheral blood monocytes shows a strong decrease in intermediate and nonclassical monocyte subsets, whatever the genetic alterations, which rapidly distinguishes a CMML from a reactive monocytosis[20,25,26]. Altered DNA methylation is another characteristic feature of CMML cells that contributes to disease phenotype[27]. Hypomethylating agents, which are commonly used to treat these patients[28], can restore a balanced hematopoiesis by demethylating leukemic cell DNA without having any impact on variant allele frequencies in the leukemic population[24].

By profiling microRNA (miRNA) expression in peripheral blood monocytes of CMML patients, we identify the down-regulation of hsa-miR-150 expression as a characteristic feature of CMML monocytes. Analysis of Mir150 knockout mice detects a cell-autonomous abnormal repartition of monocyte subsets. We show that hsa-miR-150 down-regulation is related to the hyper-methylation of a myeloid lineage-specific promoter in MIR150 gene in CMML classical monocytes. In mice and in humans, this down-regulation prevents the generation of intermediate and nonclassical monocytes through up-regulating the ten-eleven-translocation-3 (TET3) protein in classical monocytes. These results highlight a mechanism of monocyte subset differentiation that operates in both mice and humans.

## Results

**hsa-miR-150 expression is down-regulated in CMML monocytes**. To elucidate the potential role of miRNAs in the pathogenesis of CMML, we examined the expression of 851 human miRNAs in sorted peripheral blood monocytes (CD14$^+$) collected from a learning cohort of 33 severe CMML patients (Supplementary Table 1) and 5 healthy donors using microarrays. Unsupervised clustering of miRNA profiles separated CMML from control samples (Supplementary Figure 1A). Using an absolute 2-fold change and a P value <0.05 as thresholds, we observed a down-regulation of hsa-miR-150 and hsa-miR-451 while hsa-miR-494 appeared to be up-regulated in CMML compared to control samples (Fig. 1a, b and Supplementary Figure 1B). Quantitative PCR (qPCR) validation using either RNU-6B miRNA (Fig. 1c) or RPL32 or PPIA housekeeping genes (Supplementary Figure 1C) as normalizers confirmed the down-regulation of hsa-miR-150 and hsa-miR-451 in these samples, but not the up-regulation of hsa-miR-494. To further validate these results, we measured the expression of hsa-miR-150 and hsa-miR-451 in sorted peripheral blood CD14$^+$ of an independent cohort of 139 CMML patients at diagnosis (Supplementary Table 1), using CD14$^+$ sorted from 24 healthy donor blood samples as controls. While the down-regulation of hsa-miR-150 was confirmed in this cohort, we did not validate the down-regulation of hsa-miR-451 nor the up-regulation of hsa-miR-494 (Fig. 1d and Supplementary Figure 1D). Therefore, we focused our subsequent efforts in understanding the mechanisms and consequences of hsa-miR-150 down-regulation in CMML monocytes. Importantly, the expression level of hsa-miR-150 was not associated with specific disease features, including age at diagnosis, white blood cell count, absolute number of monocytes or neutrophils, hemoglobin level, platelet count (Supplementary Figure 1E), bone marrow blast cell infiltration that distinguishes CMML-0 from CMML-1 and CMML-2 (Supplementary Figure 1F)[29], and the presence of somatic mutations in TET2, SRSF2, ASXL1, RUNX1 and genes of the RAS family (NRAS and KRAS) (Supplementary Figure 1G). Together, these results indicated that hsa-miR-150 was down-regulated in peripheral blood monocytes of CMML patients, regardless of disease clinical and biological features.

**Mir150$^{-/-}$ mice show an abnormal monocyte subset repartition**. Analysis of germline Mir150$^{-/-}$ mice identified a role for mmu-miR-150 in B lymphopoiesis[30], CD8$^+$ T cell fate and function[31], natural killer (NK) cell differentiation[32,33], and megakaryocytic commitment[34]. Nevertheless, these animals demonstrate largely

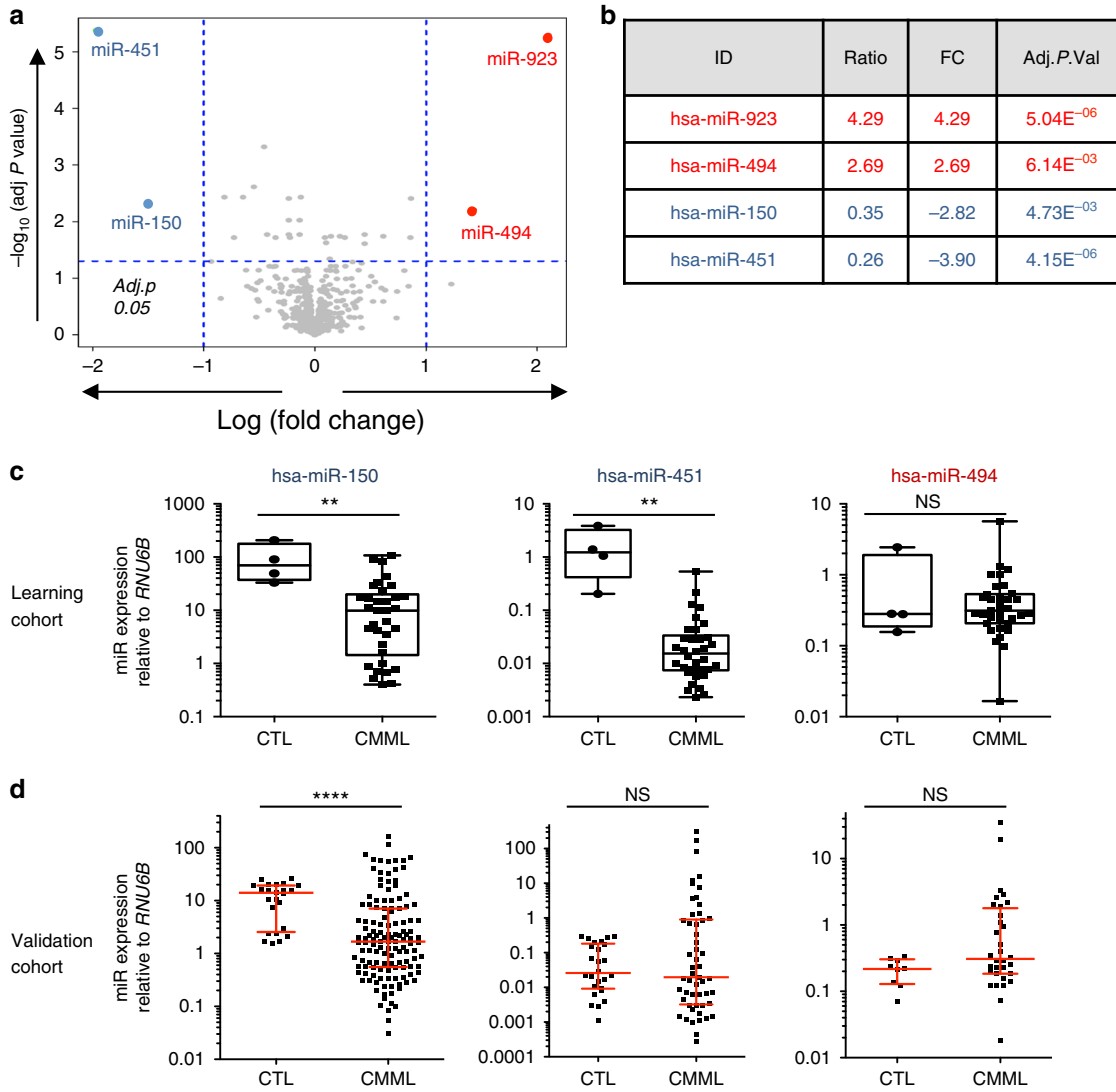

**Fig. 1** Differentially expressed miRNA from control and CMML patient-sorted CD14+. **a** Volcano plot showing the differentially expressed miRNAs in CD14+ cell-sorted monocytes from CMML ($n = 33$) and healthy donor ($n = 5$ + pool of five samples) peripheral blood samples (learning cohort), as measured using v12.0 Agilent microarray. X-axis, log 2 fold-change value; Y-axis, −log 10 (adjusted P value). **b** List of the four miRNAs identified as deregulated in the learning cohort. Of note, hsa-miR-923 up-regulation is a known artifact of the v12.0 version of these microarrays. **c** Box plot showing the relative expression of hsa-miR-150, hsa-miR-451, and hsa-miR-494 analyzed by qRT-PCR in samples of the learning cohort (center line: median; whiskers: min to max). **d** qRT-PCR analysis of the relative expression levels of hsa-miR-150 (CTL = 24; CMML = 133), hsa-miR-451 (CTL = 24, CMML = 53) and hsa-miR-494 (CTL = 9; CMML = 32) in an independent validation cohort. Red lines, median with interquartile range; Mann–Whitney test: **$P < 0.01$; ****$P < 0.0001$. NS nonsignificant

normal steady-state hematopoiesis and peripheral blood cell counts[35]. We used these mice (Supplementary Figure 2A) to determine if mmu-miR-150 deletion could affect the monocytic lineage with age, as aging is an important component of CMML pathophysiology[24]. In $Mir150^{−/−}$ animals kept aging for up to 2 years, we detected previously described changes in B cell differentiation in the spleen and a slight increase in peripheral blood leukocytes count with age (Supplementary Figure 2B), without any significant change in peripheral blood (Supplementary Figure 2C) and bone marrow (Supplementary Figure 2D) monocyte counts, nor in peripheral blood neutrophil and lymphocyte counts (Supplementary Figure 2E), nor in spleen weight. Thus, $Mir150$ deletion did not generate a CMML-like phenotype in mice. Mouse monocytes were further analyzed as side scatter$^{low}$, CD45+, B220−, CD3−, NKP46−, Ly6G−, CD115+, and CD11b+

cells and subdivided into Ly6C$^{high}$, CD43$^{low}$ classical and Ly6C$^{low}$, CD43+ nonclassical monocytes (Supplementary Figure 2F). We tested two multiparametric flow cytometry strategies, either an exclusion gating strategy (Supplementary Figure 3A) or a CD115 expression-based analysis (Supplementary Figure 3B), to quantify monocyte subsets in mouse peripheral blood and bone marrow. Both methods revealed an abnormal repartition of monocyte subsets in $Mir150^{−/−}$ mice (Supplementary Figure 3C). More specifically, we observed a significant decrease in the fraction and absolute number of Ly6C$^{low}$ cells in the bone marrow (Fig. 2a–c) and in the peripheral blood (Fig. 2d–f) of younger (<6 months) as well as older (>6 months) animals. This abnormal repartition of monocyte subsets was not related to an increased apoptosis of Ly6C$^{low}$ compared to Ly6C$^{high}$ cells, as determined by Annexin V and 7-amino-actinomycin D staining of blood and

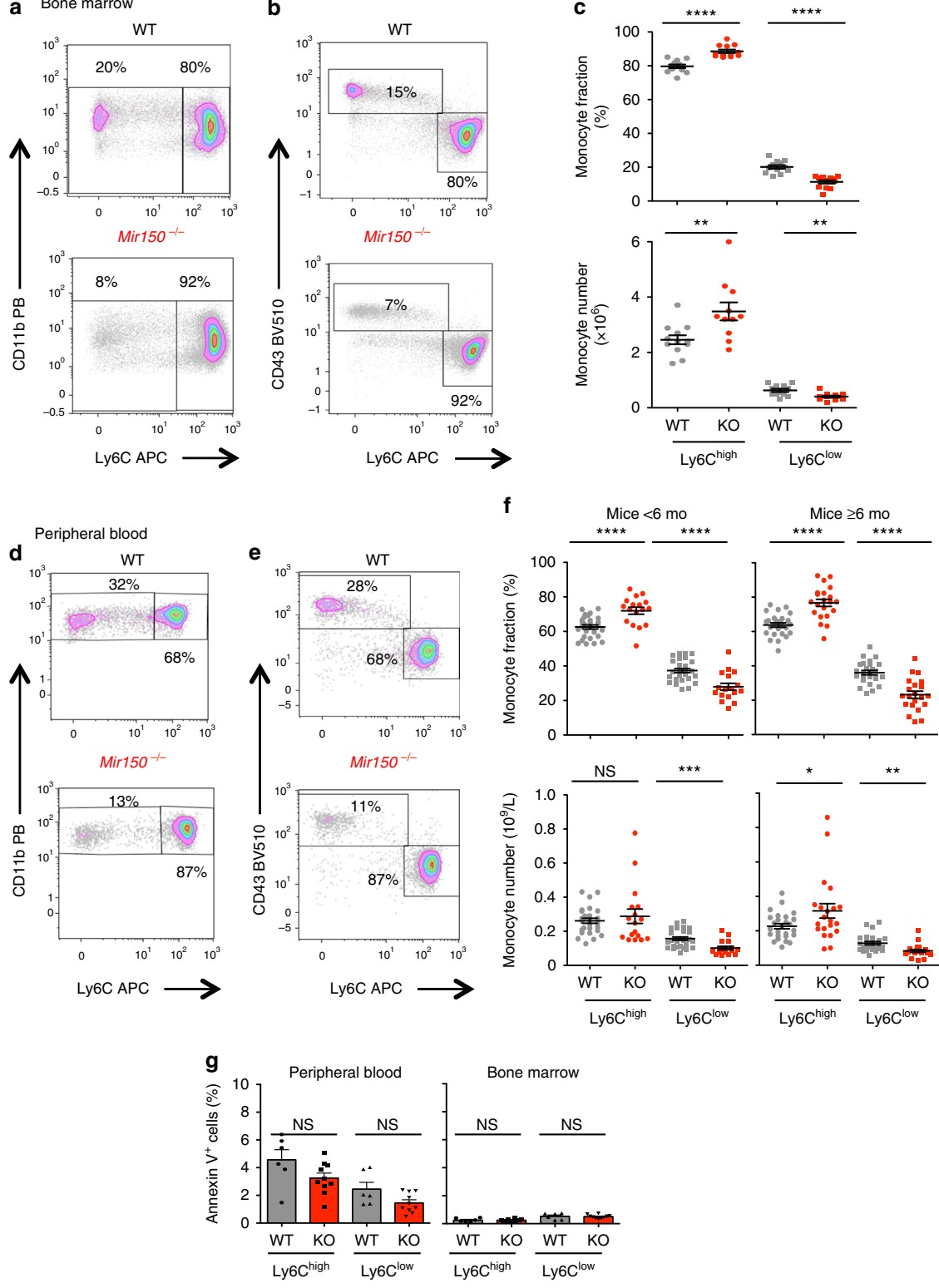

bone marrow monocytes (Fig. 2g and Supplementary Figure 4A). Also, this abnormal repartition of monocyte subsets appeared in the absence of any change in the fraction of common myeloid progenitor, granulocyte–monocyte progenitor, and megakaryocytic-erythroid progenitor (MEP) cells as well as LSK (Lin⁻,Sca⁺,Kit⁺) and LK (Lin⁻,Kit⁺) progenitor cells in the bone marrow (Supplementary Figure 4B, C). Finally, we did not detect any significant change in the mean fluorescence intensity of CD115 and CX3CR1

at the surface of Ly6C$^{low}$ $Mir150^{-/-}$ mouse monocytes (Supplementary Figure 4D).

### Cell-autonomous effect of mmu-miR-150 on subset generation.

To further explore the role of mmu-miR-150 in mouse monocyte subset generation, we transferred bone marrow cells of wild-type and $Mir150^{-/-}$ animals into lethally irradiated wild-type and $Mir150^{-/-}$-irradiated recipients. Six weeks after transplantation, the

**Fig. 2** Abnormal repartition of monocyte subsets in $Mir150^{-/-}$ (KO) mice. **a**, **b** Representative flow cytometry scatter plot of CD11b+ monocytes (**a**) or CD43+ monocytes (**b**) gated on Ly6C expression in bone marrow of a wild-type (WT) and a $Mir150^{-/-}$ mouse. **c** Flow cytometry quantification of Ly6C$^{high}$ and Ly6C$^{low}$ monocyte subsets in the bone marrow of WT and $Mir150^{-/-}$ (KO) animals (12/group) expressed as a fraction of total monocytes (upper panels) or absolute number per mouse femur (lower panels). Mean ± SEM, two-tailed unpaired $t$ test: **$P < 0.01$; ****$P < 0.0001$. **d**, **e**. Representative flow cytometry scatter plot of CD11b+ monocytes (**d**) or CD43+ monocytes (**e**) gated on Ly6C expression in the peripheral blood of a WT and $Mir150^{-/-}$ mouse. **f** Flow cytometry quantification of Ly6C$^{high}$ and Ly6C$^{low}$ monocyte subsets in the peripheral blood of young (<6 months, WT = 28, KO = 17) or older (6–26 months, WT = 26; KO = 21 KO) animals expressed as fraction of total monocytes (upper panels) or absolute number per mouse (lower panels). Mean ± SEM, two-tailed unpaired $t$ test: *$P < 0.05$; **$P < 0.01$; ***$P < 0.001$; ****$P < 0.0001$. **g** Flow cytometry measurement of the fraction of Annexin V+ cells among Ly6C$^{high}$ and Ly6C$^{low}$ monocytes in the peripheral blood and bone marrow of 6 WT (gray bars) and 10 $Mir150^{-/-}$ (red bars) animals (mean ± SEM). NS nonsignificant

repartition of monocyte subsets was normal in the blood of wild-type and $Mir150^{-/-}$ mice engrafted with wild-type bone marrow cells, whereas the fraction of peripheral blood Ly6C$^{low}$ monocytes was decreased in the blood of wild-type and $Mir150^{-/-}$ mice engrafted with $Mir150^{-/-}$ bone marrow cells (Fig. 3a, b). We then transferred various ratios (50/50, 90/10, 10/90) of bone marrow cells collected from CD45.1 wild-type and CD45.2 $Mir150^{-/-}$ mice, respectively, into lethally irradiated CD45.1 recipients (Fig. 3c). Eight weeks later, the fraction of CD45.2+ cells (among total CD45+ cells) was roughly identical to those defined in the transplanted cell mixture (Fig. 3d). The CD45.1/CD45.2 ratio in lymphocyte and monocyte populations recovered at 8 weeks post-transplantation was also similar to that defined prior to injection (Fig. 3e). Similar observations were made 12 and 40 weeks after transplantation (Supplementary Figure 5A–D). Focusing on monocyte subsets, Ly6C$^{high}$ cell fraction was higher among engrafted 45.2 $Mir150^{-/-}$ monocytes than among engrafted CD45.1 wild-type cells, and Ly6C$^{low}$ cell fraction showed the opposite repartition (Fig. 3f). These experiments suggested that the abnormal repartition of monocyte subsets observed in $Mir150^{-/-}$ mice was a cell-autonomous phenotype, with $Mir150$ gene deletion partially preventing the generation of Ly6C$^{low}$ monocytes.

To further verify that the decrease in Ly6C$^{low}$ subset was related to $Mir150$ loss, we conducted an in vivo rescue of mmu-miR-150 expression in the hematopoietic tissue. Lineage-negative (Lin$^-$) bone marrow cells from $Mir150^{-/-}$-deficient mice were transduced with a retroviral vector encoding mmu-miR-150 and green fluorescent protein (GFP), sorted to validate mmu-miR-150 overexpression compared to WT Lin$^-$ cells, and injected to lethally irradiated wild-type recipients ($n = 16$) without sorting GFP+ cells (about 25% of Lin$^-$ cells) (Fig. 3g). A variable fraction of GFP+ cells, ranging from 10 to 70% of total cells, was detected in the blood of six engrafted animals, showing the restoration of a normal monocyte subset repartition in GFP+, but not GFP$^-$, monocytes (Fig. 3h).

Since the phenotype of $Mir150^{-/-}$ mice is not the complete reduction of nonclassical monocytes otherwise observed in $Nr4a1^{-/-}$ [12,13] or $Cebpb^{-/-}$ [36,37] mice, we explored the conversion ability of Ly6C$^{high}$ by transfer experiment. A 1:1 ratio of wild-type CD45.1 and $Mir150^{-/-}$ CD45.2 Ly6C$^{high}$ monocytes were injected into transgenic mice that express GFP under the direction of the human ubiquitin C promoter[38]. After 40 h, we analyzed monocyte subsets by flow cytometry in the peripheral blood, the spleen, and the bone marrow. Dead cells and cells expressing Ly6G, CD3, B220, and NK1.1 were excluded. Ly6C$^{high}$ and Ly6C$^{low}$ subsets were quantified in CD45.1+ and CD45.2+, GFP$^-$, CD115+, CD11b+ monocyte populations. A defective conversion of Ly6C$^{high}$ into Ly6C$^{low}$ subset was observed in every studied tissue among $Mir150^{-/-}$ CD45.2 compared to wild-type CD45.1 monocytes (Fig. 3i, j and Supplementary Figure 5). Of note, the 1:1 ratio of wild-type and $Mir150^{-/-}$ monocytes at injection was conserved at 40 h,

indicating that $Mir150$ gene deletion did not promote apoptosis in this time frame.

**miR-150 is differentially expressed in monocyte subsets**. Analysis of monocyte subsets sorted from wild-type mouse peripheral blood showed a 2-fold higher expression of mmu-miR-150 in Ly6C$^{low}$ compared to Ly6C$^{high}$ monocytes (Fig. 4a and Supplementary Figure 6A). In humans, compared to sorted classical monocytes, hsa-miR-150 expression was 1.6-fold higher in intermediate monocytes and more than 10-fold higher in non-classical monocytes, respectively (Fig. 4b and Supplementary Figure 6B). We transduced human CD34+ cells with a lentiviral vector encoding GFP and either hsa-miR-150 (miR-150), or a scrambled shRNA (sh-scr) or a shRNA targeting hsa-miR-150 (sh-miR-150). We cultured these cells in a liquid medium with stem cell factor (SCF) and macrophage CSF (M-CSF) for 5 to 9 days before measuring the fraction of cells with a CD14+CD16$^-$ or a CD14+CD16+ phenotype (Fig. 4c and Supplementary Figure 6C, D). Compared to sh-SCR-transduced cells, those over-expressing hsa-miR-150 generated more CD14+CD16+ monocytes, while those transduced with sh-miR-150 (miR-150 depletion) showed the opposite effect (Fig. 4d).

**Hypermethylation of a specific $MIR150$ gene promoter in CMML**. In search for the mechanism driving hsa-miR-150 down-regulation in CMML monocytes, we first checked $MIR150$ coding sequence in sorted peripheral blood monocyte DNA collected from 49 CMML patients[24] and did not observe any mutation, nor we detected any deletion in 19q13.33 region where $MIR150$ gene is located. We then analyzed hsa-miR-150 expression in monocytes of 15 CMML patients before and after three to six cycles of the demethylating agent decitabine. While the expression of hsa-miR-150 remained low in nine non-responding patients, the six responders demonstrated an increased expression of hsa-miR-150 upon therapy (Fig. 5a), which correlated with the restoration of a normal monocyte subset repartition (Fig. 5b). These results suggested a potential epigenetic down-regulation of $MIR150$ gene expression in classical monocytes of CMML patients.

MicroRNA expression depends on lineage-specific promoters and enhancers. The 50 kb region located upstream of $MIR150$ coding sequence includes seven putative start site regions[39]. In silico analysis of chromatin immunoprecipitation-sequencing (ChIP-seq) data generated by immunoprecipitating H3K27me3, H3K4me1, H3K4me3, and H3K27ac in sorted CD3+ T lymphocytes, CD14+ monocytes, CD56+ NK cells, and CD19+ B lymphocytes indicated distinct histone mark profiles in $MIR150$ promoter regions of B, T, and NK cells compared to monocytes (Fig. 5c). H3K27ac and H3K4me3 marks suggested that the promoter region 1 (R1) was active in T, B, and NK cells, but not in monocytes in which these marks overlapped in a distinct, CpG-enriched region called region 3 (R3). CpG enrichment combined with H3K4me3 and H3K27Ac marks suggested that R3

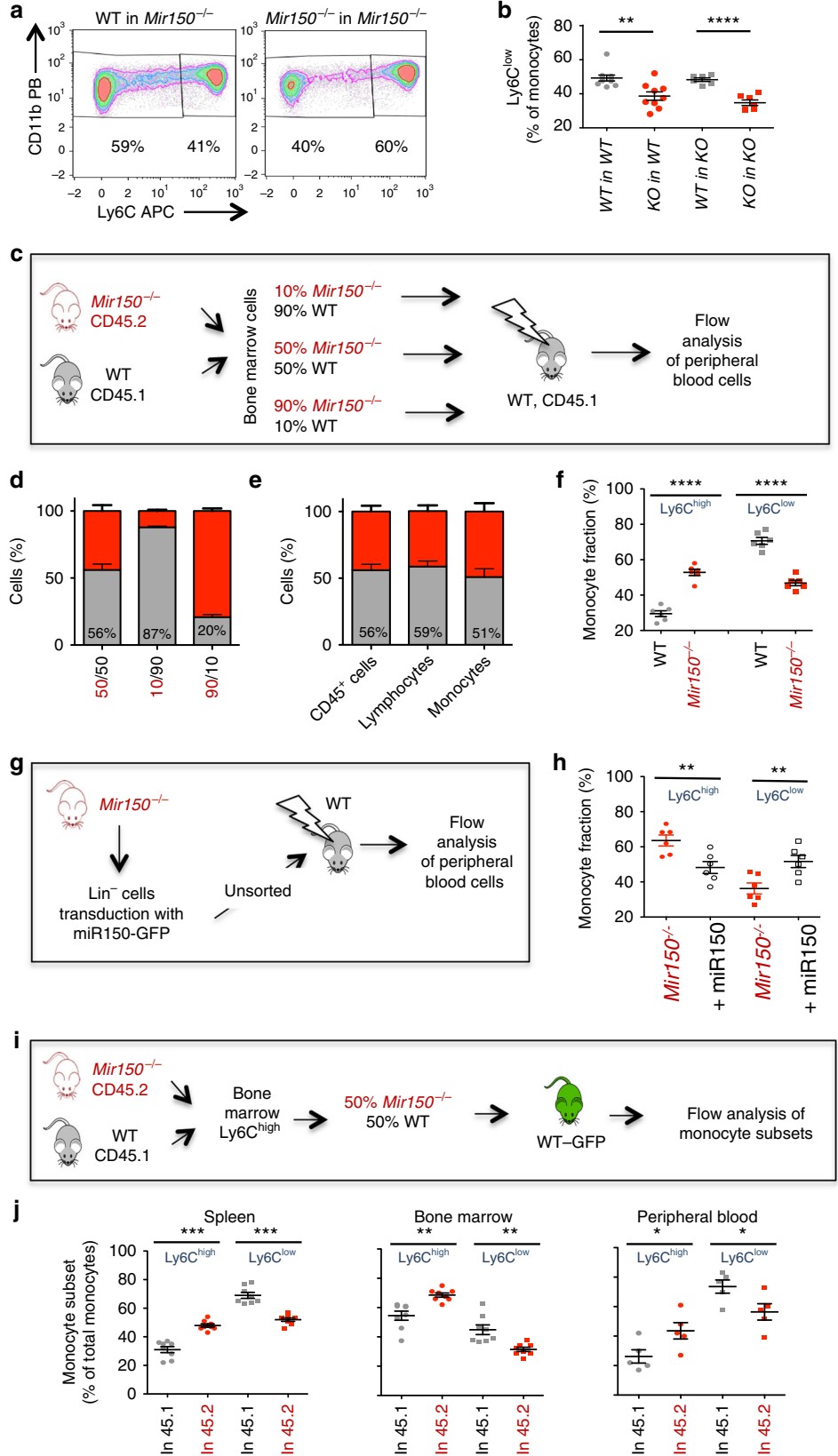

was a promoter with ongoing transcription[40,41]. GRO-cap analyses (GSE60456 accession) identified a distinct transcriptional start site (TSS) on the minus DNA strand in K562 myeloid cells and GM12878 B cells, the TSS being located on R3 in K562 cells (Supplementary Figure 7A). ChIP-seq experiments confirmed that H3K4me3 mark was located on R3 in healthy donor monocytes as well as in those collected from CMML patients (Fig. 5d; peak calling with a $P$ value of 0.05: 107,503, 119,009, and

**Fig. 3** A cell-autonomous effect of miR-150 inactivation in mouse monocyte subset generation. **a** Representative flow cytometry analysis of peripheral blood monocyte subsets in $Mir150^{-/-}$ recipient mice 6 weeks after engraftment with either wild-type (WT) or $Mir150^{-/-}$ bone marrow cells. **b** Flow cytometry quantification of Ly6C$^{low}$ monocytes in the peripheral blood of $Mir150^{-/-}$ (KO) and WT lethally irradiated recipient mice, 6 weeks after transplantation of $Mir150^{-/-}$ or WT bone marrow cells. At least six mice per group. Mean ± SEM, unpaired $t$ test, **$P < 0.01$; ****$P < 0.0001$. **c** Schema of competitive recovery assay using CD45.2$^+$ $Mir150^{-/-}$ and CD45.1$^+$ WT bone marrow cells. **d** Fraction of CD45.1$^+$ (gray) and CD45.2$^+$ (red) cells among CD45$^+$ peripheral blood cells analyzed 8 weeks after competitive transplantation at indicated ratio ($N = 6$ per group, mean ±/−SEM). **e** Expression of CD45.1 and CD45.2 in indicated cell subsets measured 8 weeks after competitive transplantation at a 50/50 ratio ($N = 6$ per group, mean ± SEM). **f** Monocyte subset repartition among CD45.1$^+$ (gray) and CD45.2$^+$ (red) cells analyzed 8 weeks after competitive transplantation of a 50/50 mixture of CD45.1$^+$ WT and CD45.2$^+$ $Mir150^{-/-}$ bone marrow cells. Mean ± SEM, paired $t$ test, ****$P < 0.0001$. **g** Schema of in vivo rescue of miR-150 expression. Lin$^-$ KO bone marrow cells were transduced with a vector expressing both mmu-miR-150 and GFP before transplantation of unsorted cells in WT animals. **h** Monocyte subset repartition of $Mir150^{-/-}$ and +miR-150 (mmu-miR-150 overexpression) cells was measured 8 weeks later in peripheral blood of chimera animals. Mean ± SEM, paired $t$ test, **$P < 0.01$. **i** Schema of monocyte transfer experiments. WT-CD45.1$^+$Ly6C$^{high}$ and $Mir150^{-/-}$-CD45.2$^+$Ly6C$^{high}$ bone marrow monocytes were mixed at a 1:1 ratio and injected into GFP-expressing mice. GFP$^-$ monocytes were analyzed 40 h later. **j** Quantification of 45.1 and 45.2 monocyte subsets in the spleen ($N = 8$), the peripheral blood ($N = 5$), and the bone marrow ($N = 8$) analyzed in three independent experiments. Mean ± SEM, paired $t$ test, *$P < 0.05$; **$P < 0.01$; ***$P < 0.001$

144,139 in control samples, and 64,730, 79,133, 134,677, and 290,419 in CMML samples) with an equal enrichment in control and CMML samples (Supplementary Figure 7B). To validate that R3 could be an active *MIR150* promoter, we used CRISP-Cas9 technology to delete a part of R3 region in U937 and K562 myeloid cell lines (Supplementary Figure 7C, D). Compared to wild-type clones, clones in which R3 has been partially deleted showed a down-regulation of hsa-miR-150 expression without any change in *FCGRT* gene expression whose TSS is close to R3 sequence, further supporting a role for R3 as a *MIR150*-specific promoter in these cells (Fig. 5e, f, Supplementary Figure 7F, G).

Bisulfite sequencing of *MIR150* R3 sequence detected an increased methylation in CMML patient monocytes compared to young and age-matched healthy donor monocytes (Fig. 6a, Supplementary Figure 8A, B). This methylation decreased in CMML patients who responded to the demethylating agent decitabine, but not in non-responders (Fig. 6b). Again, as *FCGRT* gene TSS is close to R3 sequence, we used qPCR to check that the expression of this gene remained unchanged in CMML patient compared to healthy donor monocytes (Fig. 6c and Supplementary Figure 8C). We did not detect any methylation of R1 in monocytes and in T cells of healthy donors and CMML patients, respectively, and the decreased expression of hsa-miR-150 observed in CMML patient monocytes (Fig. 6d and Supplementary Figure 8D) was not observed in their T lymphocytes (Fig. 6e and Supplementary Figure 8E). Together, these data designate R3 as an active *MIR150* promoter region in healthy monocytes and describe its methylation in CMML patient monocytes as a cause of hsa-miR-150 down-regulation.

We also compared R3 methylation in sorted classical, intermediate, and nonclassical healthy donor monocytes, which showed that the differential expression of miR-150 among these subsets did not depend on R3 methylation (Fig. 7a). ChIP-seq experiments did not detect any difference in H3K4me1, H3K27Ac, and H3K4me3 marks at R3 among monocyte subsets (Fig. 7b, c and Supplementary Figure 9A)[42]. Interestingly, while the expression of miR-150 was decreased in sorted classical and intermediate monocyte subsets collected from 10 CMML patients before any treatment and compared to healthy donor monocyte subsets, its expression was not decreased in the rare, residual nonclassical monocytes in CMML patients (Fig. 7d, Supplementary Figure 9B, C), suggesting that a fraction of leukemic cells may escape the epigenetic down-regulation of *MIR150* gene in these patients.

**TET3 is a miR-150 target that regulates monocyte subsets.** In order to identify hsa-miR-150 targets in classical monocytes,

microRNA pull-down was performed in sorted cells, 20 h after their transfection with biotinylated hsa-miR-150. Among the RNA sequences collected from four independent experiments (Supplementary Table 2) and predicted to be hsa-miR-150 targets using TargetScan algorithm, we selected *TET3* as it was also confirmed to be a potential miR-150 target using RNA-hybrid software, providing a score of −29.2 and −30.8 kcal/mol in human and mouse, respectively (Fig. 8a). *TET3* had been recently identified as a hsa-miR-150 target in acute myeloid leukemia cells[43]. These results were confirmed in U937 cells (Supplementary Figure 10A). Real-time quantitative PCR (RT-qPCR) experiments detected a decreased expression of *TET3* in monocytes generated by liquid culture of CD34$^+$ cells transduced with a hsa-miR-150-expressing lentivirus, using cells transduced with a scrambled RNA as a control (Fig. 8b and Supplementary Figure 10B). Immunoblot analysis of classical monocytes sorted from CMML peripheral blood samples showed an increased expression of TET3 protein when compared to healthy donor classical monocytes (Fig. 8c). RNA-sequencing of classical and nonclassical monocyte subsets of four healthy donors identified 2176 differentially expressed genes with an adjusted $P$ value <0.01 (Supplementary Table 3 and Supplementary Figure 10C), including *TET3* whose expression was down-regulated in nonclassical monocytes (−1.6-fold, $P = 0.001$) (Fig. 8d). Finally, an abnormal repartition of monocyte subsets, with an increase in Ly6C$^{low}$ monocytes at the expanse of Ly6C$^{high}$ cells, was detected in the blood of mice carrying inactivated *Tet3* alleles compared to wild-type littermates (Fig. 8e, f and Supplementary Figure 10D–F), without any change in the mean fluorescence intensity of CD115 and CX3CR1 at the surface of $Tet3^{-/-}$ mouse monocytes (Supplementary Figure 10G). Hence, these experiments argued for *TET3* as a target of miR-150 whose down-regulation is required for the differentiation of classical into nonclassical monocytes, in mice and in humans.

To further explore how disruption of the miR-150/TET3 axis could alter the fate of monocytes, we sequenced RNA in Ly6C$^{high}$ and Ly6C$^{low}$ monocytes collected from $Mir150^{-/-}$ and $Tet3^{-/-}$ mice and their wild-type littermates. Comparison of gene expression in Ly6C$^{high}$ and Ly6C$^{low}$ monocytes collected from the two series of wild-type littermates confirmed that *Tet3* gene expression was significantly lower in Ly6C$^{low}$ monocytes, together with *Ccr2* and *Ly6c* genes (Fig. 8g), further validating the decreased expression of *TET3* gene in nonclassical compared to classical human monocytes (Fig. 8d). Compared to wild-type cells, 127 and 78 genes were differentially expressed in $Mir150^{-/-}$ Ly6C$^{high}$ and Ly6C$^{low}$ cells, respectively (Fig. 9 and Supplementary Table 4). Of note, $Mir150^{-/-}$ gene deletion did not significantly change the expression of *Cd115*, *Cebpb*, *Nr4a1*, and

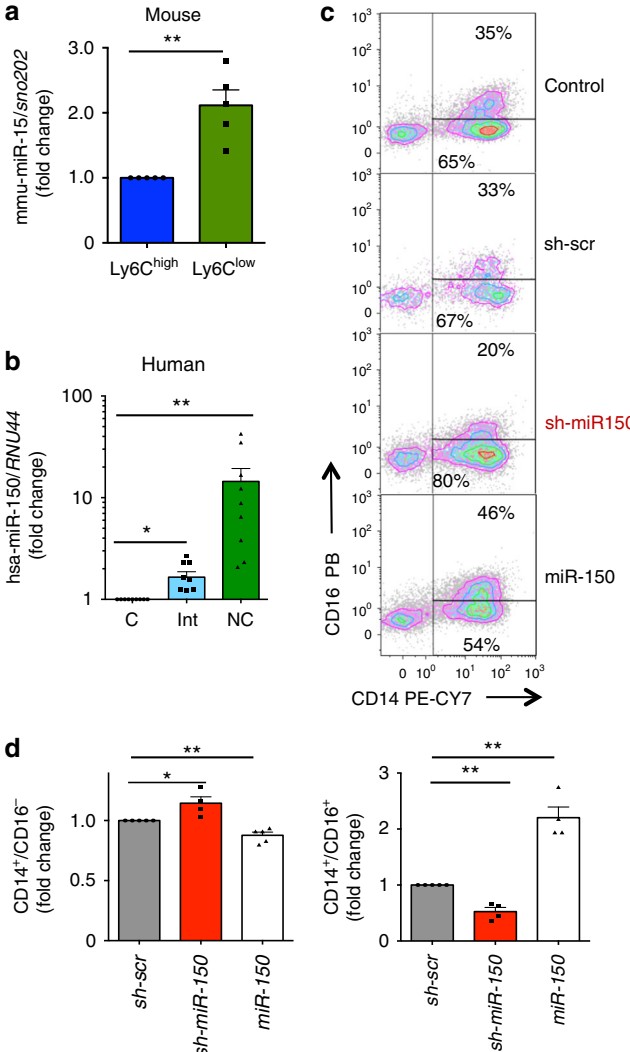

**Fig. 4** Differential expression of miR-150 in human and mouse monocyte subsets. **a** qRT-PCR analysis of mmu-miR-150 expression normalized to *sno202* in Ly6C$^{low}$ compared to Ly6C$^{high}$ monocytes sorted from wild-type mice peripheral blood. Results are expressed as fold-change relative to Ly6C$^{high}$ monocytes. Mean ± SEM of five independent experiments, paired *t* test, \*\**P* < 0.01. **b** qRT-PCR analysis of hsa-miR-150 expression normalized to *RNU-44* in peripheral blood classical (C), intermediate (Int), and nonclassical (NC) monocytes sorted from peripheral blood samples of nine healthy donor controls. Results are expressed as fold-change relative to classical monocytes. Mean ± SEM; paired *t* test, \**P* < 0.05; \*\**P* < 0.01. **c**, **d** Sorted CD34$^+$ cells were not transduced (control) or transduced with vectors expressing the GFP and a control shRNA (Sh-SCR) or a shRNA targeting hsa-miR-150 (sh-miR-150) or hsa-miR-150 (miR-150), and then cultured in liquid medium for 7 days in the presence of SCF and M-CSF before flow cytometry analysis of generated monocyte subsets, based on CD14 and CD16 expression. **c** shows a representative flow cytometry analysis. **d** shows the fraction of CD14$^+$CD16$^-$ (left panel) and CD14$^+$CD16$^+$ (right panel) generated by CD34$^+$ cells with decreased (sh-miR-150) or increased (miR-150) hsa-miR-150 level relative to controls (sh-scr). Mean ± SEM of four independent experiments, paired *t* test, \**P* < 0.05; \*\**P* < 0.01

*Tet3* genes (Supplementary Table 3). This latter result could indicate that miR-150 targets Tet3 translation rather than *Tet3* gene expression. Pathway analysis using Ingenuity software indicated most significant changes in cell migration (*z*-score −1.98, *P* value $2.76 \times 10^{-5}$) and immune cell death (*z*-score −2.13,

*P* value $6.6 \times 10^{-3}$) pathways in Ly6C$^{high}$ cells, whereas genes involved in proliferation (*z*-score −2.6, *P* value $2.88 \times 10^{-2}$) were altered in Ly6C$^{low}$ cells. Analysis of gene expression in *Tet3*$^{-/-}$ cells detected the higher number of differentially expressed genes was in *Tet3*$^{-/-}$ Ly6C$^{low}$ cells (1019 genes; Fig. 9 and Supplementary Table 4), further suggesting an important role of TET3 in the generation or functions of this monocyte subset.

## Discussion

While the diversity and versatility of circulating monocytes is well recognized, the molecular mechanisms driving the generation of minor monocyte subsets, including Ly6C$^{low}$ monocytes in mice and CD16$^+$ monocytes in humans, are only partly depicted. The present study identifies a role for miR-150-mediated regulation of *TET3* expression in the generation of these monocyte subsets. hsa-miR-150 expression in monocytes is shown to be regulated by a lineage-specific promoter whose hypermethylation in CMML cells is associated with a decrease in the fraction of circulating intermediate and nonclassical monocytes. The deletion of either *Mir150* or *Tet3* gene is associated with an abnormal repartition of monocyte subsets in mouse models.

The expression of hsa-miR-150 is deregulated in several hematopoietic malignancies[44], that is, in the most abundant miRNA detected in chronic lymphocytic leukemia B cells in which it modulates B cell receptor signaling[39] while being down-regulated in a variety of human non-Hodgkin lymphoma cells, for example, in anaplastic large-cell lymphomas in which restoration of its expression through the use of hypomethylating drugs delays disease progression in xenograft models[45]. A decreased expression of hsa-miR-150 has been also described in chronic[46] and acute[47] myeloid malignancies, for example, hsa-miR-150 maturation is inhibited and its expression is down-regulated in acute myeloid leukemias involving the mixed lineage leukemia (*MLL*) gene[47]. We detected hsa-miR-150 as the most abnormally expressed miRNA in sorted circulating monocytes of patients with a CMML, which are mostly classical monocytes, and we identified the hypermethylation of a lineage-specific promoter as the mechanism for its decreased expression. While *Mir150* deletion in mice was not sufficient to mimic all the features that define CMML phenotype, the abnormal repartition of monocyte subsets in these animals was reminiscent of that observed in CMML patients[20,26]. This observation suggested a role for miR-150 in the generation of nonclassical monocytes.

miR-150 has been involved in the development of lymphoid and myeloid lineages[44]. This microRNA plays a critical role in B cell development through directly regulating the expression level of c-Myb transcription factor[30,48], and promotes NK cell development and maturation[32]. hsa-miR-150 also drives the differentiation of myeloid progenitors toward megakaryocytes at the expense of erythroid cells[34], and regulates hematopoietic recovery after injury with chemotherapeutic drugs[35]. Among circulating monocytes, the expression of miR-150 was observed to be lower in classical than in nonclassical monocytes[49,50], with some discrepancies regarding intermediate monocytes[51] that could be related to the heterogeneity of this subset[52]. The abnormal repartition of monocyte subsets in the peripheral blood of *Mir150*$^{-/-}$ mice led us to identify a cell-autonomous function of this microRNA in the generation of Ly6C$^{low}$ monocyte subset.

Accumulating evidence supports that, in the steady state, most if not all mouse Ly6C$^{low}$ and human CD14$^-$CD16$^+$ monocytes derive from Ly6C$^{high}$ and CD14$^+$CD16$^-$ monocytes, respectively[16,53–56]. Deletion of the *Nur77/Nr4a1* gene demonstrated the requirement of the encoded transcription factor for the development and survival of Ly6C$^{low}$ monocytes[12], which was further confirmed by deletion of a unique sub-domain within the

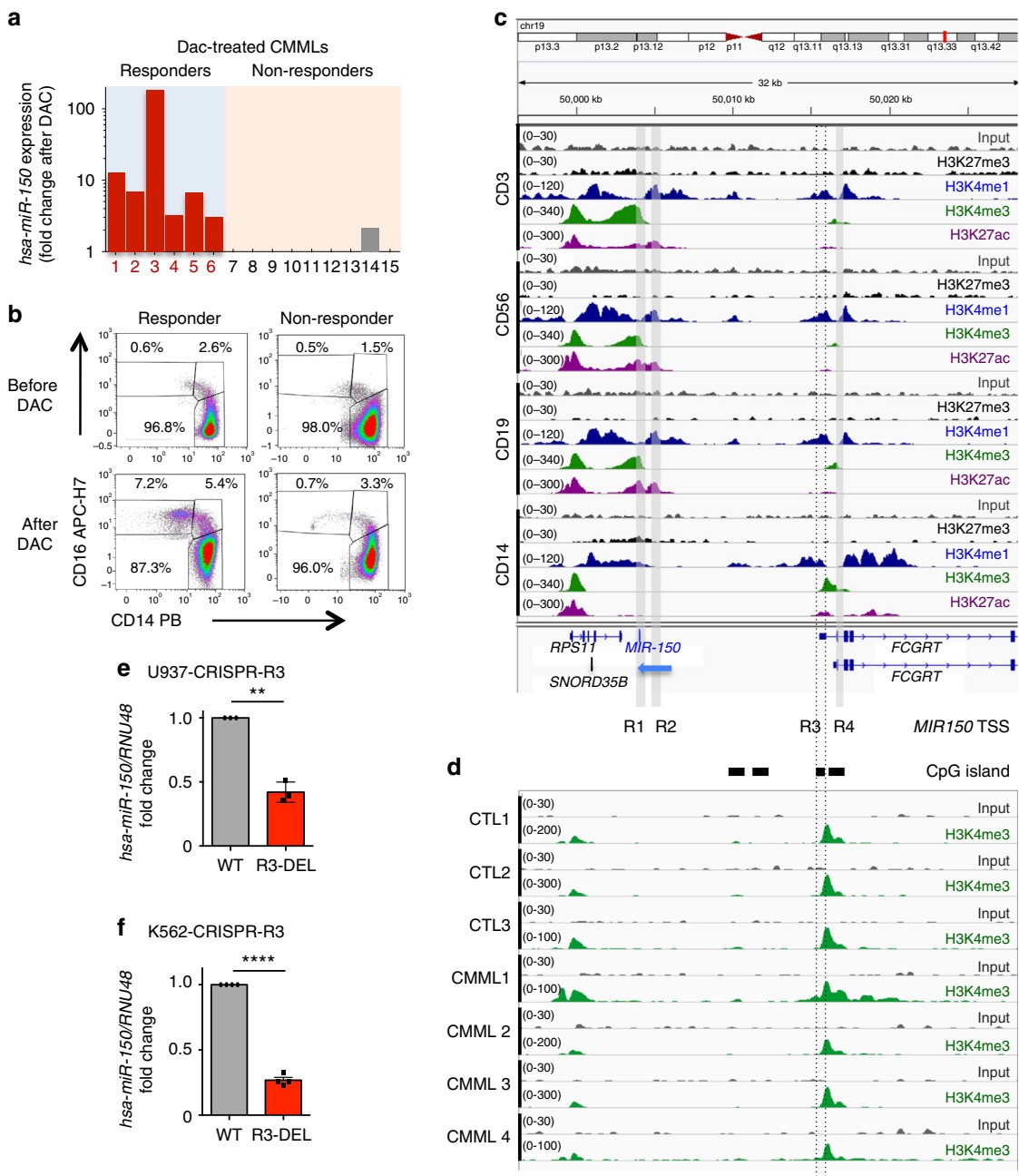

**Fig. 5** Regulation of hsa-miR-150 expression by a monocytic lineage-specific promoter. **a** Expression of hsa-miR-150 (normalized to *RNU-6B*) in monocytes of 15 CMML patients according to their response to three to six cycles of decitabine (DAC). Results are expressed as hsa-miR-150 expression fold-change after versus before treatment. **b** Representative dot plots showing the monocyte subset repartition before and after treatment with decitabine (DAC) in a responder and a non-responder CMML patient. **c** Input (gray, scale 0–30) and ChIP-seq data for H3K27me3 (black, scale 0–30), H3K4me1 (blue, scale 0–120), H3K4me3 (green, scale 0–340) and H3K27ac (purple, 0–300) obtained in human CD3+, CD19+, CD56+, and CD14+ cells were collected from www.roadmapepigenomics.org/. miR-150 promoters are indicated as R1, R2, R3, and R4. Black rectangles indicate CpG islands. *MIR150* gene (blue) is on DNA minus strand (blue arrow), while other genes (in black) are on the DNA plus strand. **d** Input (gray) and ChIP-seq of H3K4me3 (green) in sorted CD14+ cells collected from three healthy donors (CTL) and four CMML patients. Data are shown as normalized Bigwig. Scales are on the left. **e, f** Expression of hsa-miR-150 (normalized to *RNU-48*) in U937 (**e**) and K562 (**f**) clones in which R3 has been partially deleted using a CRISPR/Cas9 gene editing method. Results are expressed as hsa-miR-150 expression fold-change in R3-deleted clone (R3-DEL, red bars) relative to wild-type clone (WT, gray bars, normalized to 1). Results are mean ± SD of a minimum of three independent experiments. Paired *t* test, **P < 0.01; ****P < 0.0001

*Nr4a1* gene enhancer that is specifically required for Ly6C[low] subset development, without altering Nr4a1 expression in tissue macrophages[13].

Another transcription factor, CCAAT/enhancer-binding protein β (C/EBPβ), was recently shown to play a critical role in Ly6C[low] monocyte survival[36]. The deletion of C/EBPβ could down-regulate the expression of Csf1r (also known as Cd115 and M-CSFactor receptor) at the monocyte surface. CSF1R genetic deletion, pharmacological inhibition[57,58], and constitutive mutation[17,59] all decrease the generation of nonclassical monocyte subsets. The chemokine receptor CX3CR1, which is highly expressed on nonclassical monocytes[60], and the G-coupled

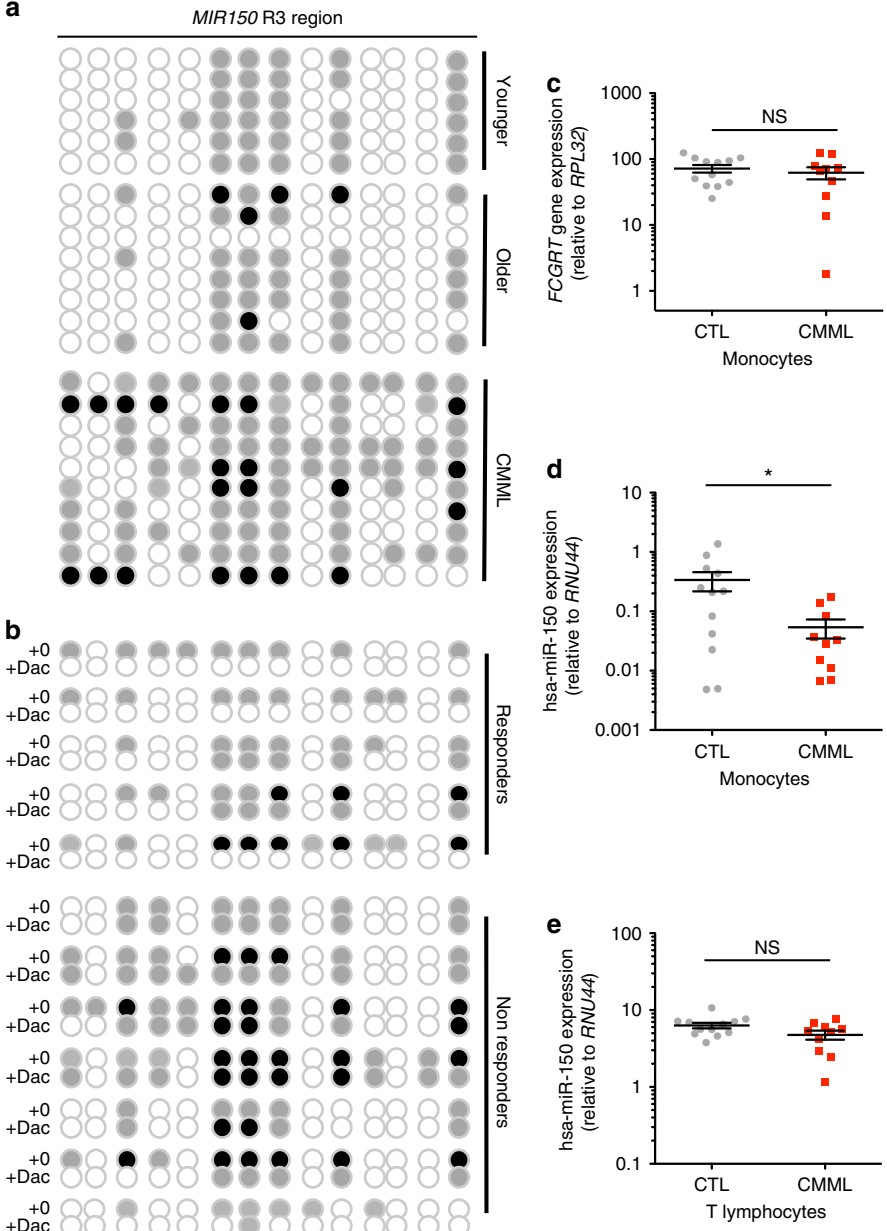

**Fig. 6** Methylation of *MIR150* region 3 in CMML classical monocytes. **a** DNA methylation of *MIR150* region 3 (R3) was analyzed in sorted CD14+ peripheral blood monocytes from six young (<65 years old) and 8 older (>65 years old) healthy donors and 10 CMML patients. White ball: no methylation; gray ball: methylated on one strand; black ball: methylated on two strands. **b** DNA methylation status of R3 was analyzed in sorted CD14+ monocytes from 12 CMML patients before (+0) and after (+Dac) decitabine treatment, including five responders and seven non-responders. White ball: no methylation; gray ball: methylated on one strand; black ball: methylated on two strands. **c–e** qRT-PCR analysis of *FCGRT* and hsa-miR-150 expression in sorted cells collected from 12 healthy donors (CTL) and 10 CMML patients. Samples are those studied by bisulfite sequencing. *FCGRT* gene expression is normalized to *RPL32* in sorted monocytes (**c**), hsa-miR-150 expression is normalized to *RNU-44* (**d**) in sorted monocytes and in CD3+ T lymphocytes (**e**). Mean ± SEM, unpaired *t* test: *P < 0.05. NS, nonsignificant

receptor for sphingosine-1 phosphate, S1PR5[61], were also suggested to regulate the fraction of circulating nonclassical monocytes. Our results identify miR-150 as an additional molecular regulator of nonclassical monocyte generation. The consequences of miR-150 down-regulation may be independent of Nr4a1, C/ EBPβ, and CSF1R, as a decrease in its expression affects the generation of nonclassical monocytes without altering their survival. Having identified the lineage-specific R3 promoter that regulates hsa-miR-150 expression in mature monocytes, a next step will be to depict the transcription factors that interact with

this sequence to promote hsa-miR-150 up-regulation along the generation of nonclassical monocytes.

While c-Myb is a direct and essential miR-150 target in B lymphocytes and megakaryocytes, we failed to identify any link between miR-150 and c-Myb expression levels in monocytes. Similarly, miR-150 could regulate effector CD8+ T cell functions independently of c-Myb[31]. In order to identify hsa-miR-150 targets in monocytes, we performed microRNA pull-down as this method allows identifying RNAs whose either stability or translation is altered by the studied microRNA. Among the predicted

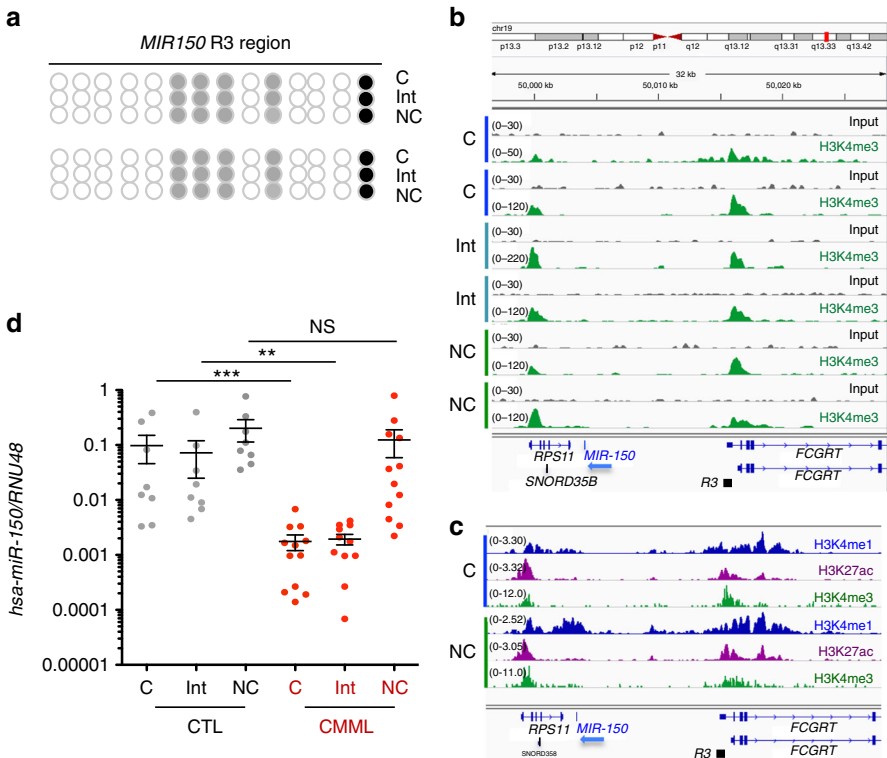

**Fig. 7** Methylation of *MIR150* region 3 in human monocytes subsets. **a** DNA methylation of *MIR150* region 3 (R3) was analyzed in classical (C), intermediate (Int), and nonclassical (NC) monocytes sorted from healthy donor peripheral blood samples. (Results of two healthy donors are shown.) White ball: no methylation; gray ball: methylated on one strand; black ball: methylated on two strands. **b**. Input and ChIP-seq of H3K4me3 in sorted classical (C), intermediate (Int), and nonclassical (NC) monocytes collected from two healthy donors. Data are shown as normalized Bigwig. R3 region is represented. **c** ChIP-seq data for H3K4me1 (blue) and H3K27ac (purple) collected from ref. [42] and for H3K4me3 (green), conducted in human classical (C) and nonclassical (Non-C) monocytes. R3 region is represented. **d** qRT-PCR analysis of hsa-miR-150 expression normalized to *RNU-48* in classical (C), intermediate (Int), and nonclassical (NC) monocytes sorted from peripheral blood samples of 8 healthy donor controls (CTL) and 12 CMML patients. Mean ± SEM; Mann–Whitney test: **$P < 0.01$; ***$P < 0.01$. NS nonsignificant

targets containing evolutionary conserved binding sites for miR-150, TET3 was of particular interest. The three TET proteins are dioxygenases that utilize Fe(II) and 2-oxoglutarate as cofactors to catalyze the oxidation of 5-methylcytosine into 5-hydroxymethylcytosine and other derivatives, thereby regulating DNA methylation[62]. Whereas *TET2* mutations are commonly detected in myeloid malignancies, especially in CMML, and mutations in *TET3* are very rare[63] and late events in clonal evolution[24]. MicroRNAs regulate the expression level of TET enzymes in a cell-specific manner[64], for example, hsa-miR-15b regulates *TET3* expression during neurogenesis[65]. *TET3* has been recently validated as a miR-150 target in acute myeloid leukemia cells[43]. The increased fraction of Ly6C$^{low}$ monocytes detected in the blood of *Tet3*$^{-/-}$ mice compared to wild-type littermates, and observations made in healthy donor and CMML patient cells, support a role for TET3 as a miR-150 target whose down-regulation is required for the optimal generation of nonclassical monocytes.

The abnormal repartition of monocyte subsets that characterizes CMML has provided a diagnostic tool in patients with a monocytosis to rapidly distinguish this myeloid malignancy from a reactive monocytosis[20]. This characteristic feature also provides new insights into circulating monocyte biology by identifying the role of a *miR-150/TET3* axis in the development of nonclassical monocyte subsets. Egression of classical monocytes from the bone marrow requires expression of the chemokine receptor CCR2[3]. These classical monocytes transit to the blood in which their half-life at steady state was demonstrated to be one day or less, in

mice[55,66] and in humans[16]. Most of these classical monocytes subsequently leave the circulation or die, whereas a small fraction of them give rise to intermediate and nonclassical monocytes and demonstrate a longer lifespan[16,55]. It remains unclear how classical monocytes are selected to differentiate into nonclassical monocytes rather than migrating into tissues. Recent investigations in mice suggested that Notch-2 signaling activated by endothelial cells was required for cell fate decision, that is, to promote classical Ly6C$^{high}$ conversion into Ly6C$^{low}$ monocytes[67]. These results suggest a scenario in which Notch-2 signaling would promote the expression of miR-150 and the modulation of downstream targets such as *TET3*. Deciphering in human miR-150 promoter regulation by key transcription factors remains to be explored.

In conclusion, our data delineate a miR-150/TET3 axis involved in the generation of nonclassical monocytes. Targeting of this pathway provides strategies to modulate monocyte subset repartition and investigate their therapeutic interest in pathological situations.

## Methods

**Healthy donor and patient samples.** Buffy coats collected from male and female healthy donors (Etablissement Français du sang, Rungis, France) and peripheral blood samples collected on ethylene diaminetetraacetic acid (EDTA) from healthy volunteers were used to sort control cells. A learning cohort of CMML (Supplementary Table 1) was made of patients enrolled between November 2008 and June 2009 in a previously reported phase 2 clinical trial (GFM-DEC-LMMC-2007-02)[68]. We also get access to sorted blood monocytes collected after three to six cycles of decitabine treatment in 15 of the patients enrolled in this trial, including six

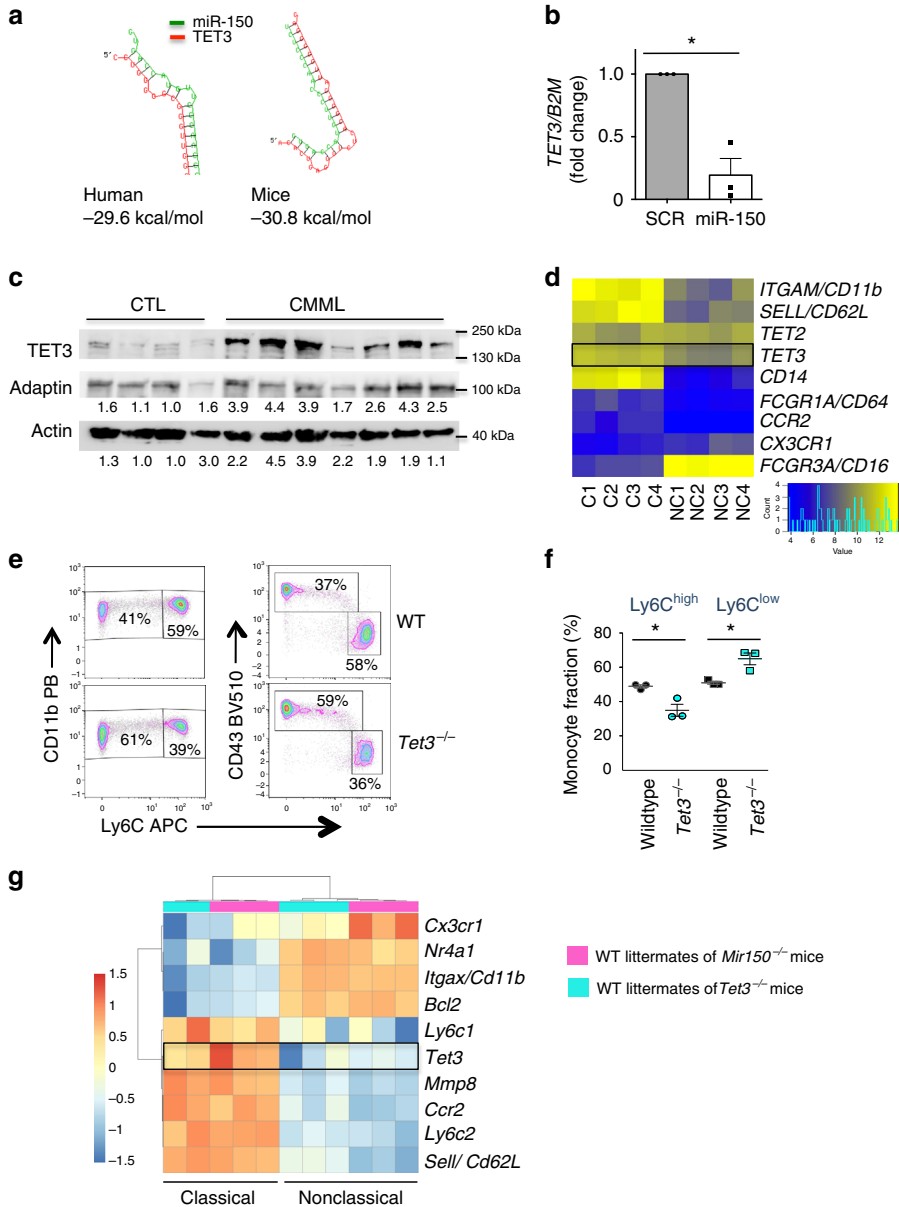

**Fig. 8** *TET3* is a miR-150 target involved in monocyte subset differentiation. **a** Hybridization analysis of 3'-UTR of *TET3* with mature sequences of miR-150 using RNA-hybrid software in human and mouse. Structures with minimum free energy are shown. **b** Expression of *TET3* (normalized to *B2M*) in monocytes generated by culture of human CD34+ cells transduced with hsa-miR-150-GFP compared to control-transduced cells (SCR). Results are expressed as *TET3* expression fold-change in hsa-miR-150-overexpressing cells relative to control (SCR, normalized to 1). Mean ± SEM of three independent experiments. Paired *t* test, *$P < 0.05$. **c** Immunoblot analysis of *TET3* expression in four healthy donor (CTL) and seven CMML patient monocytes. ADAPTIN and ACTIN are the loading controls. TET3/ADAPTIN or TET3/ACTIN ratio are shown below. Blots have been cropped for clarity. **d** Heat map of *TET2* and *TET3* gene expression, compared to a selection of differentially expressed genes in classical and nonclassical monocyte subsets as measured by RNA-seq ($N = 4$ healthy donors). The expression of genes characterizing these two subsets is shown. **e** Representative flow cytometry analysis of CD11b+ or CD43+ monocyte populations gated on Ly6C expression in peripheral blood of a wild-type (WT) and a *Tet3*−/− (KO) mice. **f** Monocyte subsets measured by flow cytometry in WT and *Tet3*−/− mice. $N = 3$, mean ± SEM, unpaired *t* test, *$P < 0.05$. One representation of three independent experiments is shown. **g** Heat map of differentially expressed genes in Ly6C^high and Ly6C^low monocyte subsets as measured by RNA-seq of two series of wild-type littermates ($N = 3$ wild-type mice for *Mir150*−/− littermates, and 2 or 3 wild-type mice for *Tet3*−/− littermates)

responders and nine non-responders. A validation cohort (Supplementary Table 1) was made of 139 newly diagnosed CMML patients whose samples were collected with informed consent following the authorization provided by the ethical committee Ile-de-France 1 (DC-2014-2091). Patients were reclassified according to the latest 2016 World Health Organization criteria[29].

**Mouse models**. All the animal experiments were approved by the Gustave Roussy ethical review board (2012-018-16-540 and 2016-104-7171). Wild-type CD45.1 and CD45.2, C57BL/6 animals expressing the GFP under human

ubiquitin C promoter[38] and *miR-150*−/− mice were purchased from Jackson Laboratories (Charles River France, L'Arbresle, France). Mice harboring *Tet3* allele with the coding sequences of exon 11 flanked by two loxP, a strategy similar that described with the Tet2 allele[69], were generated by the Plateforme Recombinaison homolog (Institut Cochin, Paris, France) and were intercrossed with mice expressing tamoxifen-inducible Cre (Cre-ER^T) transgene under control of the Scl/Tal1 promoter/enhancer. To delete Tet3-floxed alleles, tamoxifen was solubilized at 20 mg/ml in sunflower oil (Sigma-Aldrich, Saint-Quentin Fallavier, France) and 8 mg tamoxifen were administered to mice once

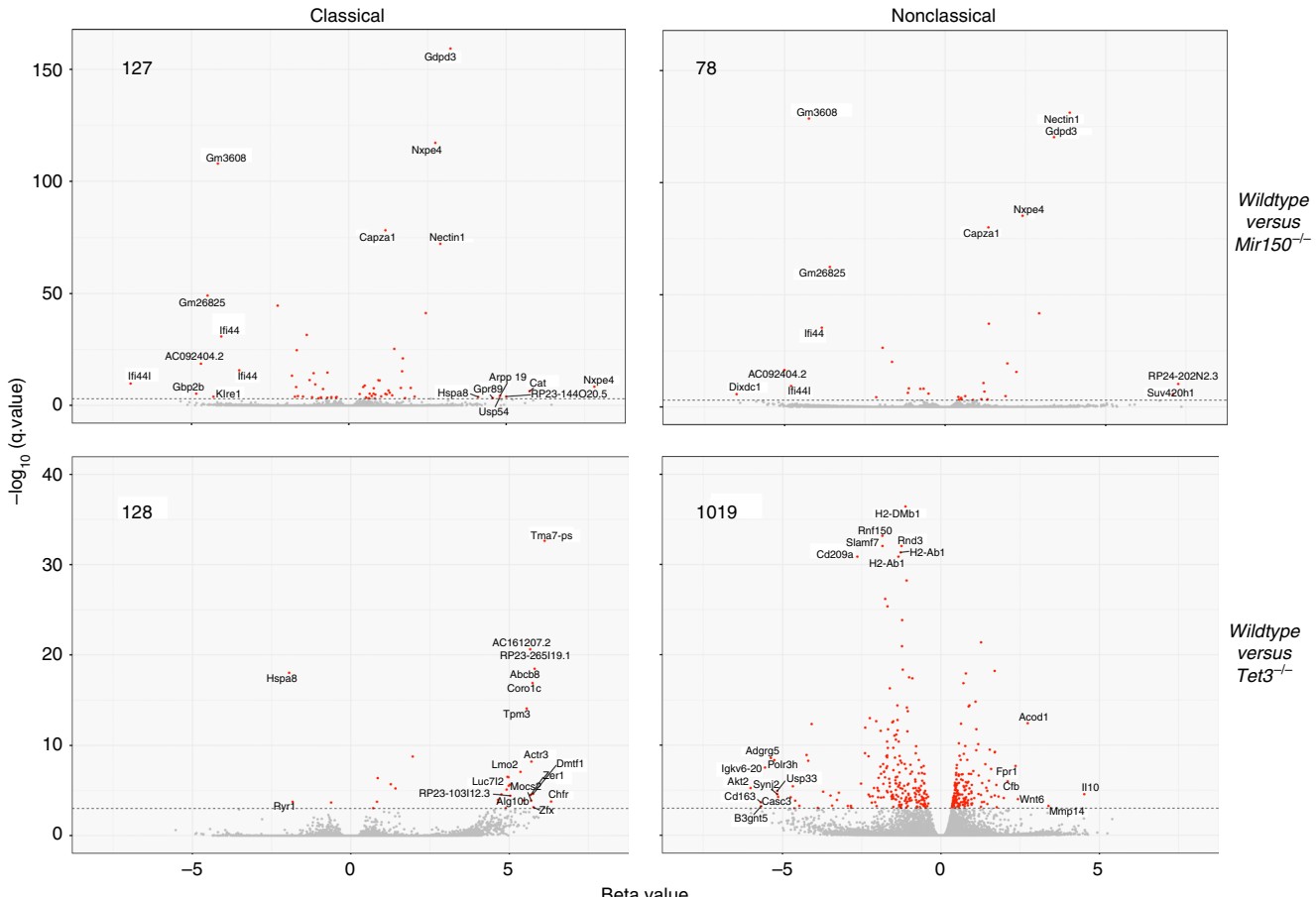

**Fig. 9** RNA-sequencing of mouse monocyte subsets. Volcano plot representation of differentially expressed genes in wild-type littermates versus either *Mir150⁻/⁻* (upper panels) or *Tet3⁻/⁻* (lower panels) monocytes subtypes. The number of differentially expressed genes (cut-off *q* value 0.001) is indicated in each panel, as well as the name of differentially deregulated genes

per day for 2 days via oral gavage. For competitive and rescue experiments, recipient mice were housed in a barrier facility under pathogen-free conditions after transplantation. Cell transfer experiments were performed in 8- to 12-week-old female mice.

**Human primary cell culture (CD34⁺) and transduction**. CD34⁺ cells were cultured 24 h at $1 \times 10^6$ cells/ml in MEM-α medium (Thermo Fisher Scientific) supplemented with 10% heat-inactivated fetal bovine serum (FBS), 1% penicillin/streptomycin and 2mM L-glutamine (Thermo Fisher), SCF (50 ng/ml), interleukin-3 (IL-3, 10 ng/ml), IL-6 (10 ng/ml), thrombopoietin (TPO, 10 ng/ml), Fms-like tyrosine kinase 3 (50 ng/ml), in a 37 °C incubator with a humidified atmosphere of 5% $CO_2$ in air. Cells were transduced at a multiplicity of infection (MOI) of 35–75 either with pRRL-EF1hsa-miR-150 or pRRL-H1-anti-miR-150 or pRRL-H1-shCTRL, sorted again 48 h after transduction using a cell sorter and anti-CD34 antibodies and cultured during 5 to 10 days in complete MEM-α with SCF (50 ng/ml) and human M-CSF (100 ng/ml). Only transduced-GFP⁺ cells were analyzed.

**Cell line culture and transfection**. The K562 cell line, originally established from the pleural effusion of a patient with chronic myeloid leukemia, and the U937 cell line, established from a histiocytic lymphoma, were purchased from ATCC and were maintained in RPMI-1640 Glutamax medium supplemented with 10% heat-inactivated FBS, 1% penicillin/streptomycin, and 2mM L-glutamine (Thermo Fisher), in a 37 °C incubator with a humidified atmosphere of 5% $CO_2$ in air. Cell lines were negative for mycoplasma contamination. Cells were transfected with lentiviral plasmids coding for the two guide RNAs (gRNA471-Cherry and gRNA289-GFP) using Lipofectamine 2000 as recommended by the manufacturer (Invitrogen, Cergy Pontoise, France), sorted after 48 h by flow cytometry for double-positive cells and cloned at one cell per well. Wild-type clones corresponding to an uncut DNA, heterozygotes and homozygous clones corresponding, respectively, to one or two alleles deleted for the R3 region.

**Antibodies and reagents**. H3K4me3 antibody used for ChIP-seq was purchased from Active Motif (La Hulpe, Belgium, no. 39159, dilution 3 μl for 100 μl). For

immunoblot experiments, we used an anti-TET3 (GTX121453, GeneTex Inc., Cliniscience, Nanterre, France, dilution 1/1000e), an anti-α-ADAPTIN (M-300, Santa Cruz Biotechnology, Cliniscience, dilution 1/1000e) and an anti-ACTIN (A5441, Sigma-Aldrich, dilution 1/5000e) antibodies. Secondary antibodies were purchased at Thermo Fisher Scientific (anti-rabbit, anti-mouse, dilution 1/20,000e). For flow cytometry, antibodies were purchased from BD Biosciences (San Diego, USA), BioLegend (San Diego, USA), and Beckman Coulter (Villepinte, France). Detailed references are in Supplementary Table 5.

**DNA and RNA preparation**. Genomic DNA and RNA were isolated using Norgen standard procedures, and total RNA from mouse monocytes was extracted using Trizol (Invitrogen, Waltham, MA, USA) and Direct-zol RNA Kit (Zymo Research, Ozyme, France). For microarray, total RNA (100 ng) was extracted using Trizol (Invitrogen, Waltham, MA, USA) from sorted CD14⁺ monocytes of 33 CMML patients and 5 healthy donors. In addition, a pool of control samples was analyzed.

**MiRNA preparation and hybridization**. RNA samples were dephosphorylated with calf intestine alkaline phosphatase (GE Healthcare Europe GmbH, Velizy Villacoublay, France), denatured with dimethyl sulfoxide, and labeled with pCp-Cy3 using T4 RNA ligase before RNA hybridization to Agilent human miRNA microarrays v3.0 (Agilent, Les Ulis, France) for 20 h at 55 °C under rotation. After washes, microarrays were scanned with an Agilent microarray scanner using high dynamic range settings before extracting data using the Agilent Feature Extraction Software (10.7.3.1). Quantile normalization was made using the normalizeBetweenArray function from R package LIMMA. Data were log 2 transformed. When several probes were spotted for a same miRNA, the median of all probes was averaged. Data were filtered according to the maximum number of missing value allowed for each miRNA (15%). Unsupervised computational analysis was performed using the dist function from R and the euclidian method as distance measure. Hierarchical clustering was obtained with the hclust function from R, using the distance matrix previously computed and the ward method. To assess differentially expressed miRNA using LIMMA package, we estimated fold changes and standard errors between the two groups by fitting a linear model for each miRNA with the *lmFit* function, and then applied an empirical Bayes smoothing to

the standard errors to the linear model. A table of the top-ranked genes was extracted from the linear model fit using the *topTable* function and results were saved as in a Volcano plot and a table file format.

**Validation of RNA expression measurements.** Validation of miRNA expression was performed by reverse transcription and real-time quantitative PCR (RT-qPCR) using Taqman microRNA Reverse Transcription Kit and detection protocol in a 7500 thermocycler (Applied Biosystems, Thermo Fisher Scientific). Validation of messenger RNA (mRNA) expression was done by reverse transcription using random hexamers and Super Script II reverse transcriptase (Thermo Fisher Scientific). RT-qPCR was performed using SYBR-Green (Applied Biosystems, Thermo Fisher Scientific). Briefly, 100–200 ng of total complementary DNA (cDNA), 50 nM (each) primers, and 1× SYBR-Green mixture were used in a total volume of 20 μL. Murine *Tet3* (#Mm01184936_g1) and *Gapdh* (#Mm99999915) Taqman probes were from Thermo Fisher. The level of expression of human miRNA (miR-150 #000473, miR-451 #478107, miR-494 #478944) was expressed relative to *sno202* #001232, *sno234* #001234, *RNU-44* #001094, *RNU-6B* #001093, *RNU-48* #001006, or *B2M* #4326319E controls (all from Applied Biosystems, Thermo Fisher Scientific), or *PPIA*, *HPRT*, or *RPL32* controls. Primer sequences: hPPIA F, 5′-GT CAACCCCACCGTGTTCTT-3′ and R, 5′-CTGCTGTCTTTGGGACCTTGT-3′, probe: AGCTCAAAGGAGACGCGGCCCA; hHPRT R, 5′-GGCAGTATAATCC AAAGATGGTCAA-3′ and F, 5′-TCAAATCCAACAAAGTCTGGCTTATAT-3′, probe: 5′-CTTGCTGGTGAAAAGGACCCCACGA-3′; mB2M F, 5′-TATATCCT GGCTCACACTGAATTC-3′ and R, 5′-TCGATCCCAGTAGACGGTCTT-3′, probe: CCCACTGAGACTGATACATACGCCTGCA. FCGRT and TET3 expression was normalized to B2M, HPRT, RPL32, and GUS for SYBR-Green experiments. SYBR primers: hB2M F, 5′-ACACAACTGTGTTCACTAGC-3′ and R, 5′-CAACTTCATCCACGTTCACC-3′ hHPRT F, 5′-GGACAGGACTGAACGTCTT GC-3′ and R, 5′-CTTGAGCACACAGAGGGCTACA-3′; hRPL32 F, 5′-TGTCCT GAATGTGGTCACCTGA-3′ and R, 5′-CTGCAGTCTCCTTGCACACCT-3′; GUS F, 5′- GAAAATATGTGGTTGGAGAGCTCATT-3′ and R, 5′-CCGAGTGA AGATCCCCTTTTTA-3′; hFCGRT F, 5′-GGAGCTTGGGTCTGGGAAA-3′ and R, 5′-GCTTCTCCTTGATCCTCAGATCTG-3′; hTET3 F, 5′-GCGCGGCATGGT ATGAA-3′ and R, 5′-ACTCGAGGTAGTCAGGGCATTCT-3′. They were expressed as the 2−^^Ct.

**Detection of gene mutations in CMML cells.** We used Ion AmpliSeq™ Custom Panel Primer Pools for myeloid genes (10 ng of genomic DNA (gDNA) per primer pool) to perform multiplex PCR with slight modifications in the generation of libraries by adding indexed paired-end adaptors (NEXTflex, Scientific) before paired-end sequencing (2 × 250 bp reads) using an Illumina MiSeq flow cell and the onboard cluster method (Illumina). The reads obtained were mapped to hg19 reference genome using bwa (v0.7.10). Varscan (v2.3.7) has been used to call the mutations with a frequency >5% in each sample. The mutations were annotated with Annovar.

**Mice analysis and bone marrow transplantation.** White blood cell count was determined using an automated counter (MS9, Schloessing Melet, France) on blood samples collected from the retro-orbital plexus in citrated tubes. Bone marrow cells were collected by flushing femurs and tibias. Spleens were excised, weighted and pushed through a 70-μm strainer. Bone marrow transplantation was performed by delivering retro-orbitaly ~5 ×10$^6$ cells from wild-type (CD45.1) or miR-150$^{−/−}$ (CD45.2) animals to lethally irradiated (9.5 Gy) recipient mice (wild-type or miR-150$^{−/−}$). For competitive transplantation experiments, lethally irradiated CD45.1 wild-type recipient mice were injected with bone marrow cells from CD45.1 wild-type donor mice mixed with cells from CD45.2 miR-150$^{−/−}$ animals. Blood was analyzed at 8, 12, and 40 weeks post transplantation. MiR-150 in vivo rescue of miR-150$^{−/−}$ expression was performed by sorting Lin$^−$ bone marrow cells from miR-150$^{−/−}$ mice with immunomagnetic beads (Biotin Mouse Lineage Depletion Cocktail, BD Biosciences, San Diego, CA, USA), culturing these cells for 24 to 48 h at 1 × 10$^6$ cells/ml in Stem Span SFEM (StemCell ref.: 5.09600% FCS, 1% penicillin/streptomycin/L-glutamine, and 100 ng/mL SCF, 100 ng/ml TPO, 50 ng/ml, IL-6, and 10 ng/ml IL-3. These cells at a density of 1.5 to 2 million cells/ml were subsequently transduced with the Megix-mmu-Mir150 at an MOI of 4–10. Forty-eight after transduction, 100,000 to 350,000 Lin$^−$ cells (of which about 25% of GFP$^+$ cells) were injected retro-orbitally to lethally irradiated wild-type recipients.

**Monocytes adoptive transfer.** Bone marrow Ly6C$^{high}$ monocytes were sorted from wild-type CD45.1$^+$ and *miR-150$^{−/−}$* CD45.2$^+$ mice and mixed in a 1:1 ratio. For each wild-type GFP$^+$ mouse recipient, 4.5 × 10$^6$ cells in 200 μl sterile phosphate-buffered saline (PBS) were injected intravenously. Recipients were sacrificed 40 h after injection and the transferred GFP$^−$ monocytes (CD3$^−$, B220$^−$, NK1.1$^−$, Ly6G$^−$, CD11b$^+$, CD115$^+$) were analyzed by flow cytometry.

**Flow cytometry analysis.** Mouse bone marrow cells were kept on ice during staining and analysis. Blood leukocytes were sorted on dextran, washed in PBS with 2 mM EDTA and kept on ice. Analysis of progenitors was made by staining 3 ×10$^6$ bone marrow cells with anti-B220, Ter119, CD11b, GR1, CD3, IL-7R PE and CD117, Sca-1, CD34, and CD16/32 antibodies. Mature cell populations were

identified by collecting two to three million of cells from blood or bone marrow samples in PBS + 2 mM EDTA + 0.5% bovine serum albumin. After blockade of Fcγ receptor for 15 min, cells were incubated with antibodies for 30 min at 4 °C. For May–Grünwald–Giemsa staining, monocyte subsets were sorted according to gating strategies in Supplementary Figure 3, centrifuged on microscope slides, dried for 1 h at room temperature, and stained with May–Grünwald–Giemsa stain, or used to collect total RNA. To detect apoptosis, cells were additionally stained with Annexin V-FITC antibody (BD Biosciences) and 7-amino-actinomycin D (Invitrogen) according to the manufacturer's protocols. Cell fluorescence was determined using a Fortessa (BD Biosciences) and analyzed with the Kaluza Software (Beckman Coulter). Fluorescence intensity was expressed as staining index, using the median and slope distributions of labeled and background cells. The absolute number of cells was calculated by multiplying the percentage of cells in a subset by the leukocyte cell count before staining. For cell analysis after transfer, SYTOX™ Blue Dead Cell Stain was used to exclude dead cells (Thermo Fisher). Human mononuclear cells were isolated from peripheral blood samples using density centrifugation Pancoll (Pan-Biotech, Dutscher, Brumath, France). Monocyte subsets repartition was quantified using CD45, CD24, CD56, CD14, and CD16 antibodies (Supplementary Table 5) as described[20]. For monocyte quantification in differentiation assays, CD163, CD71, CD14, and CD16 antibodies were used (Supplementary Table 5). Fluorescence minus one samples were made for each cytometry antibody to assure the correct gating of stained cells (Supplementary Figure 6C, D).

**Cell sorting.** Mouse monocytes were first enriched from blood or bone marrow mononuclear cells using negative selection using biotinylated anti-Ly6G, B220, CD3, NK1.1, and Ter119 antibodies (Supplementary Table 5) and magnetic beads (Streptavidin particles plus, BD Biosciences). Monocyte subsets were then sorted using Aria III, Aria-Fusion or Influx (BD Biosciences and BD Diva Software) cell sorters using Streptavidin-PE, CD11b-Pacific blue, CD115-BV605, and Ly6C-APC antibodies (Supplementary Table 5) as Lin$^−$ (CD3, B220, Ter119, Nk1.1, and Ly6G-biotin/Streptavidin-PE$^+$ cells) CD11b$^+$, CD115$^+$ monocytes, and separated according to Ly6C expression (Supplementary Figure 3B). Human CD14$^+$ cells were sorted from mononuclear cells using CD14 microbeads and the AutoMacs system (Miltenyi Biotech). Monocytes subsets were first enriched from mononuclear cells by negative selection using CD19$^−$, CD56$^−$, CD3$^−$, CD16$^−$ microbeads and the AutoMacs system (Miltenyi Biotech), and then sorted using a cell sorter and CD45, CD14, and CD16 antibodies after exclusion of CD56$^+$ cells (Supplementary Table 5). CD34$^+$ cells were sorted from human cord blood samples using magnetic beads and the AutoMacs system (Miltenyi Biotech).

**Plasmid constructs, viral production, and titration.** Murine or human gDNA was used to amplify either the murine mir-150 sequence (mmu-miR-150) using the primers mmu-miR-150-fwd: 5′-GAC AGG AAC CCC CTC CCT CAG-3′ and mmu-miR1-50-rev: 5′-AGG AAG GGA CCC AAG GCA TCC-3′, or the human hsa-miR-150 sequence (hsa-miR-150) using the primers hsa-miR-150-fwd: 5′-GTA ACG CGT CTG GAC CTG GGT ATA AGG CA-3′ and hsa-miR-150-rev: 5′-GTA GCT AGC GCA GCA GAG ATG GGA GTA CA-3′ containing a *Mlu*I and *Nhe*I restriction site, respectively. PCR fragments were cloned in TOPO2.1 (Thermo Fisher ref.: K450001) according to the manufacturer's instructions, and then transferred to the Megix retroviral backbone after an *Eco*RI digestion (mmir150) or to the pRRL lentiviral backbone (sinpRRL-PGK-GFP) after a *Mlu*I/*Nhe*I double digestion (hsa-miR-150). Human miRZip-150 anti-miR-150 microRNA construct (MZIP150-PA-1, Ozyme, France) was used to amplify the H1 promoter and the anti-miR-150 using the forward primer, 5′-CGCCTCGAGATATTTGCATGTCGC TATGTGTTCTGGG-3′ and the reverse primer, 5′-CGCCTCGAGGAATTCAAA AATCTCCCAACCCTTGTAC-3′, both with a *Xho*I restriction site. The PCR fragment was cloned into the pRRL lentiviral backbone after *Xho*I digestion and dephosphorylation. CRISPR lentiviral vector constructs: the backbone vector was constructed from the LentiCRISPR-V2puro (Addgene, #52961) to co-express *Streptococcus pyogenes* Cas9 linked to the GFP sequence by a P2A peptide driven by elongation factor-1 short. Two gRNAs (289 and 471) were selected (CRISPOR tools) to specifically delete the R3 region. In order to clone the guide sequence into the single-guide RNA scaffold, we annealed either the primer gRNA289-fwd, 5′-CAC CGT TTA ATA TCC TTC TGC GGG C-3′ with gRNA289-rev, 5′-AAA CGC CCG CAG AAG GAT ATT AAA C-3′ for the gRNA289 or the primer gRNA471-fwd, 5′-CAC CGC GAG TGA GGC AGC ATT GTA T-3′ with gRNA471-rev, 5′-AAA CAT ACA ATG CTG CCT CAC TCG C-3′ for the gRNA471. This annealing resulted in a four paired sticky end compatible with a *Bsm*BI digestion. Thus, the CRISPR backbone was digested by *Bsm*BI and ligated with the annealed gRNAs. Orientations and sequences were verified by Sanger sequencing. High-titer stocks of retroviral or lentiviral vectors were produced by transient tripartite JetPrime transfection of 293T cells with (i) the vector plasmid, (ii) an encapsidation plasmid, and (iii) a vesicular stomatitis virus envelope (VSV-g) plasmid according to the manufacturer's instructions (Polyplus, #114-07). Forty-eight hours after transfection, supernatants were concentrated by ultracentrifugation. Vector titers were determined by transduction of serial dilutions of the concentrated supernatant in 293T cells and measurement of GFP expression. Plasmid constructs and U937 transduction for GFP reporter assay are described in Supplementary methods.

**ChIP-sequencing**. Cells were cross-linked with addition of 1% formaldehyde to the culture medium for 10 min at room temperature with agitation. Fixation was stopped by the addition of 125 mM glycine during 5 min at room temperature with agitation before washing samples twice in ice-cold PBS and adding sodium dodecyl sulfate lysis buffer (Millipore, 10 μl per $1 \times 10^6$ cells) supplemented with 1% protease inhibitor cocktail (Active Motif). Samples were incubated 15 min on ice before storage at −80 °C. Cross-linked DNA were sonicated using $12 \times 24$ mm$^2$ round bottom tube, 10 min at 40 W (Covaris S220, Woodingdean, UK). ChIP was carried out using ChIP-it Express Kit according to the manufacturer's instruction (Active Motif). Enriched DNA from ChIP and Input DNA fragments were end-repaired, extended with an "A" base on the 3' end, ligated with indexed paired-end adaptors (NEXTflex, Bioo Scientific, Proteigene, Saint Marcel, France) using the Bravo Platform (Agilent, Les Ulis, France), size selected after four cycles of PCR with AMPure XP beads (Beckman Coulter, Villepinte, France), and amplified by PCR for 10 more cycles. Fifty-cycle single-end sequencing was performed using HiSeq 2000 (Illumina) for controls and CMML CD14$^+$ monocytes, or fifty-cycle single-end sequencing with NovaSeq 6000 (Illumina) for control monocyte subsets, in order to reach at least 20 millions reads per sample. Reads were aligned into human genome hg19 with BWA aln (v0.7.5a) and peak calling assessed using MACS 2.0 with a $q$ value cut-off of 0.05 for the histone mark H3K4me3. Annotation has been done with HOMER (v4.7.2), with a $P$ value of 0.01. Integrative Genomics Viewer (IGV 2.1) was used for representation. Normalized bigwig files have been computed using bamcomverage from deeptools 3.1.2, with a bin size of 50, the number of read per bin normalized using RPKM methods, reads extension set to 250 bp, and a smooth length windows of 150 bp. Reads with the same orientation and start have been considered only once. From normalized bigwig files, the distribution of signal for each mark and the corresponding heatmap have been computed using computeMatrix and plotHeatmap from deeptools 3.1.2, using TSS from Human genome annotation file (genecode.v19.annotation.gft) as the reference point.

**Bisulfite DNA sequencing**. Two hundred nanograms of total gDNA were modified by bisulfite treatment according to the manufacturer's instructions (Methyl-Detector, Active Motif). Converted R3 *miR-150* promoters were identified by PCR using primers MIR150convR3 F2, 5'-GGTATTTTTTTAAAAAATTTTTGTAGG TG-3' andR, 5'-CTCTATTACCAATCTAAAATACAATAATACAATC-3', and direct sequencing reaction was performed using standard conditions (Applied Biosystems).

**Protein extraction and immunoblotting**. Sorted monocytes were lysed in 2× Laemmli with 0.1 M dithiothreitol, boiled 10 min at 95 °C, and sonicated to collect total proteins, which were separated on polyacrylamide gel and transferred to nitrocellulose membrane (Thermo Fisher Scientific). Membranes were blocked with 5% bovine serum albumin in PBS, with 0.1% Tween-20 (Sigma-Aldrich) for 40 min at room temperature, incubated overnight at 4 °C with the primary antibodies, washed in PBS-0.1% Tween-20, incubated further with horse radish peroxidase-conjugated secondary antibody (400 ng/ml) for 1 h at room temperature and washed again before analysis using Immobilon Western Chemiluminescent HPR Substrate system (Millipore, Molsheim, France). The chemiluminescent emission was registered by imageQuant LAS 4000 camera (GE Healthcare Life Science, Vélizy, France). Uncropped scans of key immunoblots are shown in Supplementary Figure 11.

**miR-150 pull-down**. Monocytes were sorted with Monocyte Isolation Kit II (Miltenyi Biotech) before resuspending $5 \times 10^6$ cells in 100 μl nucleofector solution with 2 pM hsa-miR-150 or miR-39p, a control mi-RNA from *Caenorhabditis elegans* (miRcury LNA microRNA mimic, Exiqon). Cells were electroporated in duplicate (Lonza, Amboise, France), cultured overnight at $10^6$/ml in RPMI-1640 Glutamax medium (Thermo Fisher Scientific) supplemented with 10% heat-inactivated FBS, 1% penicillin/streptomycin, and 2 mM L-glutamine (Thermo Fisher Scientific) for 20 h, and lysed. MicroRNA pull-down was then performed as described[70]. Briefly, monocytes were washed twice with PBS and resuspended in a lysis buffer. After 20 min on ice, lysates were spun at $5000 \times g$ for 5 min at 4 °C. The supernatant was incubated with magnetic beads (Dynabeads, Life Technologies) for 4 h on a rotator. Beads were washed five times in the lysis buffer and resuspended in 100 μl of lysis buffer. RNAs were isolated with 500 μl of Trizol LS (Life Technologies). RNA obtained were purified on Amicon ultra-0.5 centrifugal filter devices (Millipore). Four independent experiments were made on distinct buffy coats. Two hundred picograms of concentrated RNA were first-strand synthesized and tailed using SMARTScribe reverse transcriptase (SMARTer Universal Low Input RNA, Clontech). The cDNA strand was then template switched and extended by SMARTScribe reverse transcriptase. Double-stranded cDNA was amplified by 25 cycles of PCR using Advantage 2 PCR Kit (Clontech), purified by AMPure beads (were end-repaired, extended with an "A" base on the 3' end, ligated with indexed paired-end adaptors (NEXTflex, Bioo Scientific) using the Bravo Platform (Agilent) and amplified by PCR for 10 cycles. Indexed libraries were sequenced in an Illumina MiSeq flow cell using the onboard cluster method, as paired-end sequencing (2 × 250 bp reads) (Illumina, San Diego, CA, USA). The reads were adapter and quality trimmed with Trimmomatic (v0.32). Cleaned reads

were mapped to hg19 reference genome with Tophat2 (v2.0.14)/Bowtie2 (v2.1.0) with the following parameters: --read-realign-edit-dist 0 --library-type fr-unstranded. The regions covered were identified with Bedtools (v2.17.0) and annotated with Homer (v4.7.2).

**RNA-sequencing**. RNA integrity (RNA integrity score ≥7.0) was checked on the Agilent 2100 Bioanalyzer (Agilent) and quantity was determined using Qubit (Invitrogen). SureSelect Automated Strand Specific RNA Library Preparation Kit was used according to the manufacturer's instructions with the Bravo Platform. Briefly, 100 ng of total RNA sample was used for poly-A mRNA selection using oligo(dT) beads and subjected to thermal mRNA fragmentation. The fragmented mRNA samples were subjected to cDNA synthesis and were further converted into double-stranded DNA using the reagents supplied in the kit, and the resulting double-stranded DNA was used for library preparation. The final libraries were sequenced on an Hiseq 2000 for human samples and on NovaSeq 6000 for mice samples (Illumina) in paired-end 100 bp mode in order to reach at least 30 millions reads per sample at Gustave Roussy.

For human samples analysis, raw reads were mapped to hg19 genome with Tophat2 (v2.0.14)/Bowtie2 (v2.1.0) with the following parameters: --read-realign-edit-dist 0 --library-type fr-firststrand --mate-inner-dist 50. Expression analysis was performed by counting the number of reads per gene using HTSeq (0.5.4p5) with the following parameters: -m union -s reverse -a 10 and GENECODE gene annotation (v24lift37). DESeq2 (v1.10.1) package was used for differential gene expression analysis.

For mouse sample analysis, Fastq files quality have been analyzed with FastQC (v0.11.7) and aggregated with MultiQC (v1.5). The quantification was performed on Gencode mouse M18 (GRCm38p6) transcriptome and comprehensive gene annotation, with Salmon (v0.10.2). The index was build with the default k-mer length of 31, with the genecode flag on, the perfect hash option, and all 1569 sequence duplicates within the genome were kept. The quantification was done with the default expected maximization algorithm, verified through 100 boostrap rounds, sequence-specific bias correction, fragment GC-bias correction, and automatic library detection parameter. Clustered heatmaps were performs from normalized counts with pheatmap, an R package, using Pearson's coefficient as distance metric for rows and column and the Ward.D2 method for the clustering. Volcano plot were built using ggplot package. The differential analysis was performed with Sleuth (v0.29.0), on data converted by wasabi (v0.2).

**Statistical analysis**. Student's $t$ test were performed using the Prism software.

## Data availability

Datasets are available in the ArrayExpress database at EMBL-EBI (www.ebi.ac.uk/arrayexpress). H3K4me3 CHIP-Seq comparing CMML and healthy donor monocytes profiles are referenced as E-MTAB-6689 and those comparing human classical, intermediate, or nonclassical monocytes profiles are referenced as E-MTAB-7290. RNA-sequencing comparing CMML and healthy donor monocytes expression profiles are referenced as E-MTAB-6712 and those RNA-sequencing comparing classical or nonclassical monocytes expression profiles between wild-type and *MiR-150*$^{-/-}$ or wild-type and *Tet3*$^{-/-}$ mice are referenced as E-MTAB-7291. GRO-cap analyses obtained in a human myeloid (K562) and a human lymphoid (GM12878) cell line (Supplementary Figure 7A) were collected from www.ncbi.nlm.nih.gov/geo/ (GSE60456 accession). ChIP-seq data for H3K27me3, H3K4me1, H3K4me3, and H3K27ac obtained in human CD3$^+$, CD19$^+$, CD56$^+$, and CD14$^+$ cells (Fig. 5C) were collected from www.roadmapepigenomics.org/. GSM1027287_UW.CD19_Primary_Cells.H3K27ac.RO_01701.Histone.DS21712.wig: GSM1027288_UW.CD56_Primary_Cells.H3K27ac.RO_01701.Histone.DS21716.wig: GSM1027290_UW.CD19_Primary_Cells.H3K27me3.RO_01679.Histone.DS22585.wig: GSM1027291_UW.CD56_Primary_Cells.H3K27me3.RO_01679.Histone.DS22594.wig: GSM1027296_UW.CD19_Primary_Cells.H3K4me1.RO_01679.Histone.DS22584.wig: GSM1027297_UW.CD56_Primary_Cells.H3K4me1.RO_01679.Histone.DS22593.wig: GSM1027300_UW.CD19_Primary_Cells.H3K4me3.RO_01701.Histone.DS21711.wig: GSM1027301_UW.CD56_Primary_Cells.H3K4me3.RO_01701.Histone.DS21715.wig: GSM1027304_UW.CD19_Primary_Cells.Input.RO_01701.Histone.DS21628.wig: GSM1027305_UW.CD56_Primary_Cells.Input.RO_01701.Histone.DS21629.wig: GSM1058764_UW.CD3_Primary_Cells.H3K27ac.RO_01701.Histone.DS21704.wig: GSM1058778_UW.CD3_Primary_Cells.H3K4me1.RO_01679.Histone.DS22561.wig: GSM1058782_UW.CD3_Primary_Cells.H3K4me3.RO_01701.Histone.DS21703.wig : GSM1058789_UW.CD3_Primary_Cells.Input.RO_01701.Histone.DS21626.wig: GSM1102782_UW.CD14_Primary_Cells.H3K27ac.RO_01721.Histone.DS22926.wig: GSM1102785_UW.CD14_Primary_Cells.H3K27me3.RO_01721.Histone.DS22405.wig: GSM1102787_UW.CD3_Primary_Cells.H3K27me3.RO_01679.Histone.DS22927.wig: GSM1102793_UW.CD14_Primary_Cells.H3K4me1.RO_01721.Histone.DS22403.wig: GSM1102797_UW.CD14_Primary_Cells.H3K4me3.RO_01701.Histone.DS21707.wig: GSM1102807_UW.CD14_Primary_Cells.Input.RO_01701.Histone.DS21627.wig

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

## Acknowledgements

Our research was supported by the Ligue Nationale Contre le Cancer (équipes labellisées LIGUE 2017: E.S. and O.A.B.) and grants from Institut National du Cancer (INCa) to E.S. (TRANSLA13-157; PRT-K16-057; PRT-K16-067) and to O.A.B. (PLBIO16-151), Association Laurette Fugain (ALF 2017/04), Gustave Roussy taxe d'apprentissage 2009 and 2016 (to R.I. and L.B., respectively), and ANR-granted Molecular Medicine in Oncology and PACRI programs. The authors are indebted to Jeffie Lafosse for collecting clinical and biological annotations on CMML patients, and physicians and patients of the Groupe Francophone des Myélodysplasies for providing samples. We are also grateful to Enguerran Mouly and Veronique Della Valle for the establishment and maintenance of the *Tet3*$^{-/-}$ mice and to Françoise Porteu, Muriel Gaudry, Jean-Luc Villeval, and Fawzia Louache for scientific discussions.

## Author contributions

D.S.-B. designed, performed, analyzed the experiments, and wrote the manuscript, J.R. performed cloning experiments and transduction experiments, H.G., C.L. and F.D. provided assistance for animal experiments, L.B. performed RNA-sequencing experiments on monocyte subsets, M.M. sorted monocytes and performed RNA extraction from patient samples, M.D. and G.M. performed bioinformatic analyses, M.B. prepared and performed libraries for sequencing, A.C. and C.D. performed qPCR experiments, R. I., C.W., N.C. provided patient samples, O.W.-B. provided assistance for mice flow cytometry analyses, O.B. provided advices and suggestions regarding TET3 experiments, N.D. supervised molecular biology experiments, E.S. supervised the project and wrote the manuscript.

## Additional information

**Competing interests:** The authors declare no competing interests.

