## [Peer Review File · Nature Communications]

Reviewers' comments:

Reviewer #1 (Remarks to the Author):

The study by Selimoglu-Buet et al. entitled 'A mir-150/TET3 pathway regulates the generation of mouse and human non-classical monocyte subset' addresses the molecular pathways involved in monocytic developmental abnormalities observed in patients with chronic myelomonocytic leukemia (CMML). First, the authors identified miR-150 as one of the strongest down-regulated miRNAs in monocytes isolated from patients with CMML. In mice, knockout of this non-coding RNA resulted in a 'comparable' monocyte phenotype as observed in CMML patients, meaning a 50% reduction of non-classical monocytes. This phenotype was cell intrinsic and overexpression of miR-150 rescued the phenotype in mice and increased human CD14⁺CD16⁺ monocyte generation in vitro. The demethylating agent decitabine increased the fraction of non-classical monocytes in responding patients, suggesting an increased methylation of the miR-150 regulatory region, which indeed could be demonstrated in CMML patients. Finally, the authors identified by micro-RNA pull-down experiments TET3 as a possible target and indeed, in classical monocytes from CMML patients elevated TET3 protein levels could be verified. Also a Tet3^{-/-} mouse model underlines the importance of Tet3 in monocyte subset distribution, as these mice had increased frequencies of non-classical monocytes.

All together, the here presented results are of high quality and the study is throughout all experiments well conducted. The authors used state-of-the-art technology and performed all necessary control experiments. However, in terms of the question 'what happens to the non-classical monocyte subset under these conditions?', the reviewer has a few suggestions:

Major points:

1. The observed phenotype in miR-150^{-/-} mice is a 50% reduction of circulating Ly6Cl^o monocytes. As published by the authors in a previous paper (Selimoglu-Buet et al., 2015), CMML patients were characterized by an almost complete absence of non-classical monocytes (>90% reduction) and the discrepancy should be clearly mentioned in the text. Anyway, as the phenotype is not a complete reduction of non-classical monocytes (as it is the case for nr4a1^{-/-} or cebpb^{-/-} mice), the Ly6Chi might be in part blocked in their conversion abilities. This can be tested by transfer experiments of Ly6Chi monocytes. Co-injection of CD45.1/1 WT Ly6Chi and CD45.2/2 miR150^{-/-} or Tet3^{-/-} Ly6Chi monocytes into CD45.1/2 hosts and analysis of their conversion pattern over time should be performed.

2. As mentioned before, even though the authors nicely demonstrated a monocytic role for the miR150/Tet3 axis, the exact fate of monocytes is unclear. Are the monocytes stuck in the classical monocyte stage or recruited to other sites? The classical monocytes might also be activated and thereby circumvent conversion. It would strengthen the paper if the authors could perform some transcriptional analysis (e.g. RNA-Seq) for classical and non-classical monocyte subsets isolated from miR150^{-/-}, Tet3^{-/-} and littermate controls. A comparison to classical human CMML monocyte transcription profiles would give further inside into the fate of monocytes under these circumstances.

Minor points:

1. Even though the authors state in the text that the R3 region does not affect FCGRT expression in K562 and U937 cells (page 15), this data should be shown in the supplement.

2. As CD115 and CX3CR1 control Ly6Cl^o monocyte survival, are there any differences in the mean fluorescence intensities of these proteins between WT and miR150⁻ or Tet3-deficient non-classical monocytes?

3. The Sci/Tal1-CreERT2 mouse needs further validation and explanation. Is the PCR in S8C performed on monocytes? The last band does not look like a complete knockout mouse. Can the authors perform real-time PCR on sorted monocytes for Tet3 to show the degree of Tet3 decrease? When after tamoxifen treatment was the experiment performed? Control animals were treated with tamoxifen as well?

4. Please indicate antibody clones.

5. Please avoid statements like 'dramatic increase' (discussion page 20).

Reviewer #2 (Remarks to the Author):

The manuscript by Selimoglu-Buet et al. describes the influence of miR-150 on the formation of non-classical monocytes, a subpopulation that arises from normal monocytes and exhibit main functions in wound healing. Reduced amounts of non-classical monocytes can be found in several diseases, including CMML. Here, miR-150 is remarkably decreased in CD14+ PB monocytes of CMML patients compared to healthy donors. This decreased expression seems to be due to promoter methylation. In addition, the authors showed that a miR-150 knockout in mice leads to reduced numbers of non-classical monocytes and increased numbers of classical monocytes. Finally, TET3 seems to be an important target of miR-150 during the development of non-classical monocytes.

The study is interesting, the experiments are over all well performed and the manuscript is mainly well written. In addition, the reviewer clearly appreciates the experimental effort using a miR-150-/- mouse model and primary CD34+ hematopoietic cells. However, there are some general issues regarding the scientific impact of the presented findings:

(1) Most of the presented data are mainly descriptive. What is the rationale in miR-150 depletion for the function of non-classical monocytes within CMML or healthy cells? Is there any biological consequence? The miR-150 KO effects are quite mild and do not completely prevent nc monocyte formation. The reduction from 20% to 10% of CD14+ cells illustrates that miR-150 is not essential for nc monocyte formation, but reduces the total amount of cells. Are the remaining 10% nc monocytes (without any miR-150) biologically different from miR-150 wildtype nc monocytes?
(2) The reviewer is really honoring the huge experimental effort within the in vivo animal model and the time-consuming retransplantation and miR-150 rescue investigations. However, the authors should have put more focus on the general disease and/or biological relevance. For instance, does miR-150 overexpression and the resulting formation of more non-classical monocytes have any consequence for the disease biology of CMML?

Specific major pitfalls:

(1) How can the authors explain the huge variances of the miRNA expression levels between their Learning and Validation cohort (especially for miR-451...)? And does the presented data fit to a recent publication by Zawada et al. (Immunobiology 2018)? On page 18, the authors state that miR-150 has been shown to be lower expressed in classical than in non-classical monocytes. They cite Zawada et al. But within this publication, non-classical miR-150 expression is higher than in intermediate monocytes and similar to that of classical monocytes. Regarding the huge variation within the authors initial data from Figure 1, the conclusions are somehow difficult. This should have been at least discussed.

(2) In Figure 3H: It is unclear why the authors just compared GFP+ to GFP- cells. The transduction itself can have a huge impact on cell behavior. A scrambled miRNA or GFP only coding vector should have been used (like for the human system in figure 4)

(3) In Figure 4: The results from Figure 4C do not match with the bar graphs in Figure 4D. The FACS plots show 80% CD14+/CD16- cells after miR-150 overexpression (the highest percentage of all conditions). In contrast, they show 20% CD16+/CD14+ in the miR-150 overexpression condition. This is the lowest percentage of all conditions. Within the bar graphs, it is exactly the

opposite. How can this be explained?

(4) In Figure 5A. Responders show accelerated miR-150 expression in CD14⁺ cells. How were the FACS panels gated? Does it include all CD14^{low} nc-monocytes? As the authors showed an increased expression of miR-150 in nc-monocytes compared to normal monocytes, these results are not surprising and might be independent of the demethylative activity of the DAC treatment. Moreover, it might just verify the results of Figure 4b. If you have more nc monocytes, miR-150 expression is higher. It would have been more interesting, if endogenous miR-150 expression in CD14⁺ cells before DAC treatment could predict responsiveness.

(5) Figure 6: How is the methylation status of the R3-MIR150 promoter in nc monocytes compared to normal monocytes (either in healthy cells or in CMML, taken into account that nc monocytes are decreased in CMML)?

(6) A luciferase assay with mutations within the specific binding site(s) is necessary to prove the direct interaction of miR-150 and TET3 3'UTR in monocytic cells (e.g. U937).

Reviewer #3 (Remarks to the Author):

Review Nat COMM:

The manuscript by Selimoglu-Buet et al, proposes a regulatory role for miR-150 in generation of non-classical monocytes via targeting TET3 mRNA with a focus on CMML.

The main concept of the manuscript is to assess the role of miR-150 in the proportional change of classical and non-classical monocytes in CMML patients. The study does not contain a lot of novelty compared to the publication of the same group in the manuscript of 2016 entitled "MIR150 Is Involved in the Monocyte Subset Differentiation and Its Down-Regulation Leads to Classical Monocyte Accumulation in Chronic Myelomonocytic Leukemia". The manuscript is a follow up study with additive experiments.

Comments:

In the first section of the results, the authors report on the down-regulation of miR-150 expression in peripheral blood monocytes from CMML patients as compared to healthy individuals. The comparison was carried out on CD14⁺ monocyte only, however the down-regulation of miR-150 may result from the different proportion of the monocyte sub-types (CD14⁺/CD16⁺) rather than expression of miR-150 itself. It is necessary that the authors analyze the expression of miR-150 in different subtypes: intermediate and the non-classical monocytes between CMML and healthy subjects.

The main phenotype of miR-150 'de-regulation' is the proportional shift (repartitioning) between classical vs non-classical monocytes in CMML vs healthy. This proportional shift could simply be due to a significant influx of immature CD14⁺/CD16⁻ monocytes in the blood or alternatively, the CD16⁺ cells migration into tissues. Both scenarios would result in repartitioning of the sub-types but not necessarily the differentiation interruption. What is the proportion of CD34⁺ cells in the blood of CMML patients compared to healthy individuals? The authors should elaborate on this?

In the knock-down experiment using shRNA against miR-150, specifically on fig4-C, how come that sh-MIR150 treated CD34⁺ cells resulted in higher CD16⁺ cells as compared to NT. The same applies to miR150 over-expression plot! Intuitively, one would expect the opposite? Can the authors explain this?

In the ChIPseq experiments in fig5-C, the authors showed that different histone marks are having different coverage across different cell types including CD14⁺ monocytes, T cells, NK cells and B cells. The authors did not provide the same data for the CD16⁺ monocytes as one of the main cells being under investigation. The authors should perform chip-seq data for CD16⁺/CD14⁻ cells?

This would show the epigenetic makeup of the poised R3 region in CD16+ cells.

In the ChIPseq experiments in fig5-C-D, what are the normalized peak counts/numbers for each histone mark in healthy vs CMML CD14+ monocytes? The authors should provide numbers and statistical analysis. Are the histone marks differentially occupied in R3 region across replicates, and in CMML vs healthy individuals? The authors should show the epigenetic makeup of promoter of FCGRT gene if R3 is in fact miR-150 related promoter? The authors also need to provide the input tracks of each mark related to each sample to document the actual signal-to-noise ratio for different histone marks.

In the knock-out experiment fig5-E-F, how do the authors explain that miR-150 is expressed after deletion of R3 as it is suggested that it contains the miR-150 promoter? The presence of H3K4me1 and H3K27ac over R3 suggests that it could very well be the active enhancer rather than a promoter in monocytes. The authors should proof with additional experiments that R3 is the promoter.

10- To strengthen the suggested direct interaction of miR-150/TET3 pathway in monocyte repartitioning, the authors should perform a Tet3 rescue experiment in Tet3^{-/-} cells to see if that can change the proportional phenotype back to normal as was done for miR-150 rescue experiment.

Minor point:

Page 16, line 8-11 " We cultured these cells in liquid medium with stem cell factor and M-CSF for 5 to 9 days, before analyzing, after having excluded CD71+ Or CD163+ macrophages, the percentage of CD14+CD16⁻⁻⁻ and CD14+CD16⁺ Generated monocytes " is unclear and needs to be rephrased.

The figures are in general not immediately informative. For example, changing the color of H3K27ac track in figure 5-C and correcting the upside-down x axis labels in figure S3-S7.

Our response to Reviewer #1

Comment from the reviewer: *The study by Selimoglu-Buet et al. entitled 'A mir-150/TET3 pathway regulates the generation of mouse and human non-classical monocyte subset' addresses the molecular pathways involved in monocytic developmental abnormalities observed in patients with chronic myelomonocytic leukemia (CMML). First, the authors identified miR-150 as one of the strongest down-regulated miRNAs in monocytes isolated from patients with CMML. In mice, knockout of this non-coding RNA resulted in a 'comparable' monocyte phenotype as observed in CMML patients, meaning a 50% reduction of non-classical monocytes. This phenotype was cell intrinsic and overexpression of miR-150 rescued the phenotype in mice and increased human CD14⁺CD16⁺ monocyte generation in vitro. The demethylating agent decitabine increased the fraction of non-classical monocytes in responding patients, suggesting an increased methylation of the miR-150 regulatory region, which indeed could be demonstrated in CMML patients. Finally, the authors identified by micro-RNA pull-down experiments TET3 as a possible target and indeed, in classical monocytes from CMML patients elevated TET3 protein levels could be verified. Also a tet3^{-/-} mouse model underlines the importance of Tet3 in monocyte subset distribution, as these mice had increased frequencies of non-classical monocytes. All together, the here presented results are of high quality and the study is throughout all experiments well conducted. The authors used state-of-the-art technology and performed all necessary control experiments. However, in terms of the question 'what happens to the non-classical monocyte subset under these conditions?', the reviewer has a few suggestions*

Our response: We thank the reviewer for these positive comments and his/her suggestions that we carefully considered, as detailed below

Comment from the reviewer, major point 1. *The observed phenotype in mir150^{-/-} mice is a 50% reduction of circulating Ly6C^{Lo} monocytes. As published by the authors in a previous paper (Selimoglu-Buet et al., 2015), CMML patients were characterized by an almost complete absence of non-classical monocytes (>90% reduction) and the discrepancy should be clearly mentioned in the text. Anyway, as the phenotype is not a complete reduction of non-classical monocytes (as it is the case for Nr4a1^{-/-} or Cebpb^{-/-} mice), the Ly6C^{hi} might be in part blocked in their conversion abilities. This can be tested by transfer experiments of Ly6C^{hi} monocytes. Co-injection of CD45.1/1 WT Ly6C^{high} and CD45.2/2 Mir150^{-/-} or Tet3^{-/-} Ly6C^{high} monocytes into CD45.1/2 hosts and analysis of their conversion pattern over time should be performed.*

Our response: We agree that miR150 epigenetic down-regulation may not be the only mechanism involved in the decreased fraction of peripheral blood non-classical monocytes observed in CMML. The disease has been kind of a front door to explore the role of miR150 in the generation of monocyte minor subsets. This is now better stated in the revised version of the manuscript (see below) to introduce the suggested transfer experiments in order to determine if Ly6C^{hi} cells were partially blocked in their conversion abilities.

As recipients, rather than CD45.1/2 animals, we used Green Fluorescent Protein (GFP)-expressing transgenic mice, which express GFP under the direction of the human ubiquitin C promoter in all tissues examined, including hematopoietic cell [C57BL/6-Tg(UBC-GFP)30Scha/J, Jackson laboratories] (described in Schaefer BC, et al. Cell Immunol. 2001;214:110-22). Of note, only 5 blood samples could be analyzed as the number of cells recovered in this tissue was very low in three animals.

Modifications: These results are shown in the **new figure 3, panels I and J**. They are described at the end the “mmu-miR-150 has a cell-autonomous effect on mouse monocyte subset generation” chapter, as follows: “Since the phenotype of *Mir150*^{-/-} mice is not the complete reduction of non-classical monocytes observed in *Nr4a1*^{-/-12,13} or *Cebpb*^{-/-36,37} mice, we explored the conversion ability of Ly6C^{high} by transfer experiment. A 1 to 1 ratio of wild-type CD45.1 and *Mir150*^{-/-} CD45.2 Ly6C^{high} monocytes were injected into transgenic mice that express Green Fluorescent Protein (GFP) under the direction of the human ubiquitin C promoter.³⁸ Forty hours later, we analyzed monocyte subsets by flow cytometry in the peripheral blood, the spleen and the bone marrow. Dead cells and cells expressing Ly6G, CD3, B220 and NK1.1 were excluded. Ly6C^{high} and Ly6C^{low} subsets were quantified in CD45.1⁺ and CD45.2⁺, GFP⁻, CD115⁺, CD11b⁺ monocyte populations. A defective conversion of Ly6C^{high} into Ly6C^{low} subset was observed in every studied tissue among *Mir150*^{-/-} CD45.2 compared to wild-type CD45.1 monocytes (**Figure 3I, 3J, and S5**). Of note, the 1/1 ratio of wild-type and *Mir150*^{-/-} monocytes at injection was conserved at 40 hours, indicating that *Mir150* gene deletion did not promote apoptosis in this time frame.

Comment from the reviewer, major point 2. *As mentioned before, even though the authors nicely demonstrated a monocytic role for the miR150/Tet3 axis, the exact fate of monocytes is unclear. Are the monocytes stuck in the classical monocyte stage or recruited to other sites? The classical monocytes might also be activated and thereby circumvent conversion. It would strengthen the paper if the authors could perform some transcriptional analysis (e.g. RNA-Seq) for classical and non-classical monocyte subsets isolated from miR150^{-/-}, Tet3^{-/-} and littermate controls. A comparison to classical human CMML monocyte transcription profiles would give further inside into the fate of monocytes under these circumstances.*

Our response: Thank you for this important suggestion. We have performed the suggested RNA Seq analyses of Ly6C^{high} and Ly6C^{low} monocytes collected from *miR150*^{-/-} mice, *Tet3*^{-/-} mice, and their respective WT littermates.

Modifications: “To further explore how disruption of the miR-150/TET3 axis could alter the fate of monocytes, we sequenced RNA in Ly6C^{high} and Ly6C^{low} monocytes collected from *Mir150*^{-/-} and *Tet3*^{-/-} mice and their wildtype littermates. Comparison of gene expression in Ly6C^{high} and Ly6C^{low} monocytes collected from the two series of wildtype littermates confirmed that *Tet3* gene expression was significantly lower in Ly6C^{low} monocytes, together with *Ccr2* and *Ly6c* genes (**Figure 8G**), further validating the decreased expression of *TET3* gene measured in non-classical compared to classical human monocytes (**Figure 8D**). Compared to wild-type cells, 127 and 78 genes were differentially expressed in *Mir150*^{-/-} Ly6C^{high} and Ly6C^{low} cells, respectively (**Figure 9; Table S3**). Of note, *Mir150*^{-/-} gene deletion did not significantly change the expression of *Cd115*, *Cebpa* and *Nr4a1* genes (**Table S3**). Pathway analysis using Ingenuity software indicated most significant changes in cell migration (z score -1.98, pval 2.76.10⁻⁵) and immune cell death (z score -2.13, pval 6.6.10⁻³) pathways in Ly6C^{high} cells, whereas genes involved in proliferation (z score -2.6, pval 2.88.10⁻²) were altered in Ly6C^{low} cells. Analysis of gene expression in *Tet3*^{-/-} cells detected the higher number of differentially expressed genes was in *Tet3*^{-/-} Ly6C^{low} cells (1,019 genes, **Figure 9**), (**Table S3**) further suggesting an important role of TET3 in the generation or functions of this monocyte subset.”

Comment from the reviewer, minor point 1. Even though the authors state in the text that the R3 region does not affect *FCGRT* expression in K562 and U937 cells (page 15), this data should be shown in the supplement.

Our response: This data has been added to **Supplemental Figure 7, panels E and F**, with *B2M* gene as a normalizer.

Modifications: Changes in **Supplemental Figure 7, panels E and F**, are as follows: **E, F.** Expression of hsa-miR-150 or *FCGRT* in U937 (**E**) or K562 (**F**) clones in which *MIR150* R3 has been partially deleted. Results are expressed as expression fold change between wildtype clone (WT, =1, grey bars) and R3-deleted clone (R3-DEL, Red bars). hsa-miR-150 expression is normalized to that of *RNU44* or *HPRT*. *FCGRT* expression is normalized to *B2M* (similar results with *RPL32* or *HPRT* as normalizers). Results are mean +/-SD of a minimum of 3 independent experiments. Paired t test, * P<0.05; ** P<0.01; *** P<0.001.

To the reviewer: Using another normalizer (*RPL32*), changes in *FCGRT* gene expression were still non-significant:

Comment from the reviewer, minor point 2. As *CD115* and *CX3CR1* control *Ly6C^{lo}* monocyte survival, are there any differences in the mean fluorescence intensities of these proteins between WT and *mir150*^{-/-} or *Tet3*-deficient non-classical monocytes?

Our response: No, we did not detect any significant change in the staining index of these two proteins at the surface of mouse monocytes, which is now shown as **figure S4D** for peripheral blood *mir150*^{-/-} and **figure S10F** for *Tet3*^{-/-} *Ly6C^{low}* monocyte.

Modifications:

“Finally, we did not detect any change in the mean fluorescence intensity of *CD115* and *CX3CR1* at the surface of *mir150*^{-/-} mouse monocytes (**Figure S4D**)”.

“Finally, an abnormal repartition of monocyte subsets, with an increase in *Ly6C^{low}* monocytes at the expense of *Ly6C^{high}* cells, was detected in the blood of mice carrying inactivated *Tet3* alleles compared to wildtype littermates (**Figure 8E, 8F and S10C, D, E**), without any change in the mean fluorescence intensity of *CD115* and *CX3CR1* at the surface of *Tet3*^{-/-} mouse monocytes (**Figure S10F**).”

Comment from the reviewer, minor point 3. The *Scl/Tal1-CreERT2* mouse needs further validation and explanation. Is the PCR in *S8C* performed on monocytes? The last band does not look like a complete knockout mouse. Can the authors perform real-time PCR on sorted monocytes for

Tet3 to show the degree of Tet3 decrease? When after tamoxifen treatment was the experiment performed? Control animals were treated with tamoxifen as well?

Our response: The mouse model was generated by introducing LoxP sites into intron 10 and the 3'UTR of exon 11 (E11) before crossing with Scl-Cre animals. Both Cre⁻ and Cre⁺ mice were treated with Tamoxifen when 2-month old.

Modifications:

The construction is now shown on **Figure S10C**.

We provide a new multiplex PCR analysis performed on monocytes sorted from *Tet3*-deleted (5 to 8) and control (# 1 to 4) animals, either 7 months (# 3,4 and # 7,8) or 16 months (# 1,2 and # 5,6) after tamoxifen treatment (**Figure S10D**).

We performed qRT-PCR analyses of *Tet3* gene expression in monocytes sorted from *Tet3*-deleted (N = 4) and control (N = 5) animals. Normalizer, *gapdh* gene. **** $P < 0.0001$ (unpaired Student's t-test) (**Figure S10E**).

Comment from the reviewer, minor point 4. *Please indicate antibody clones.*

Modifications: Antibody clones are now provided as **Table S4**.

Comment from the reviewer, minor point 5. *Please avoid statements like 'dramatic increase' (discussion page 20).*

Modifications: We have removed this statement.

Our response to Reviewer #2

Comment from the reviewer: *The manuscript by Selimoglu-Buet et al. describes the influence of miR-150 on the formation of non-classical monocytes, a subpopulation that arises from normal monocytes and exhibit main functions in wound healing. Reduced amounts of non-classical monocytes can be found in several diseases, including CMML. Here, miR-150 is remarkably decreased in CD14+ PB monocytes of CMML patients compared to healthy donors. This decreased expression seems to be due to promoter methylation. In addition, the authors showed that a miR-150 knockout in mice leads to reduced numbers of non-classical monocytes and increased numbers of classical monocytes. Finally, TET3 seems to be an important target of miR-150 during the development of non-classical monocytes. The study is interesting, the experiments are over all well performed and the manuscript is mainly well written. In addition, the reviewer clearly appreciates the experimental effort using a miR-150^{-/-} mouse model and primary CD34+ hematopoietic cells. However, there are some general issues regarding the scientific impact of the presented findings:*

Our response: We thank the reviewer for these positive comments and for having detected some mistakes in the manuscript. We paid attention to every raised issue and fixed our errors, as described below.

Comment from the reviewer: *(1) Most of the presented data are mainly descriptive.*

Our response: We questioned the mechanisms that control monocyte subset generation, taking advantage of the abnormal repartition of monocyte subsets in CMML. In this context, we are more than descriptive as we performed a number of experiments in which miR150 expression is experimentally modified, both in human and mouse cells, and in which *tet3* gene is deleted in mice. We also performed CRISPR/Cas9 deletion of the R3 region in *MIR150* gene and explored the functional regulation of its expression in untreated and treated samples. Finally, we performed pull-down experiments to identify miR150 targets, validated miR150 interaction with TET3, and added to this revised version the results of transfer experiments (see **Figure 3, panels I and J**).

Comment from the reviewer: *(1) Most of the presented data are mainly descriptive. What is the rationale in miR-150 depletion for the function of non-classical monocytes within CMML or healthy cells? Is there any biological consequence? The miR-150 KO effects are quite mild and do not completely prevent nc monocyte formation. The reduction from 20% to 10% of CD14+ cells illustrates that miR-150 is not essential for nc monocyte formation, but reduces the total amount of cells. Are the remaining 10% nc monocytes (without any miR-150) biologically different from miR-150 wildtype nc monocytes?*

We agree that miR150 loss does not completely block the generation of Ly6C^{low} monocytes in mice. Our data suggest that the epigenetic down-regulation of *MIR150* expression contributes to the decrease in the fraction of non-classical monocytes observed in CMML. It may not be the only mechanism involved. This is better stated in the revised version of the manuscript as an introduction to transfer experiments.

Modifications: "Since the phenotype of *Mir150*^{-/-} mice is not the complete reduction of non-classical monocytes observed in *Nr4a1*^{-/-12,13} or *Cebpb*^{-/-36,37} mice, we explored the conversion ability of Ly6C^{high} by transfer experiment."

To determine if miR150 deletion has biological effects on monocyte functions, following also a suggestion by reviewer 1, we have performed RNA-Seq analyses of Ly6C^{high} and Ly6C^{low} monocytes collected from *miR150*^{-/-} mice and their WT littermates. These results have been added to the revised version of the manuscript

Modifications: “To further explore how disruption of the miR-150/TET3 axis could alter the fate of monocytes, we sequenced RNA in Ly6C^{high} and Ly6C^{low} monocytes collected from *Mir150*^{-/-} and *Tet3*^{-/-} mice and their wildtype littermates. Comparison of gene expression in Ly6C^{high} and Ly6C^{low} monocytes collected from the two series of wildtype littermates confirmed that *Tet3* gene expression was significantly lower in Ly6C^{low} monocytes, together with *Ccr2* and *Ly6c* genes (**Figure 8G**), further validating the decreased expression of *TET3* gene measured in non-classical compared to classical human monocytes (**Figure 8D**). Compared to wild-type cells, 127 and 78 genes were differentially expressed in *Mir150*^{-/-} Ly6C^{high} and Ly6C^{low} cells, respectively (**Figure 9; Table S3**). Of note, *Mir150*^{-/-} gene deletion did not significantly change the expression of *Cd115*, *Cebpa* and *Nr4a1* genes (**Table S3**). Pathway analysis using Ingenuity software indicated most significant changes in cell migration (z score -1.98, pval 2.76.10⁻⁵) and immune cell death (z score -2.13, pval 6.6.10⁻³) pathways in Ly6C^{high} cells, whereas genes involved in proliferation (z score -2.6, pval 2.88.10⁻²) were altered in Ly6C^{low} cells. Analysis of gene expression in *Tet3*^{-/-} cells detected the higher number of differentially expressed genes was in *Tet3*^{-/-} Ly6C^{low} cells (1,019 genes, **Figure 9**), (**Table S3**) further suggesting an important role of TET3 in the generation or functions of this monocyte subset.”

Comment from the reviewer: (2) *The reviewer is really honoring the huge experimental effort within the in vivo animal model and the time-consuming retransplantation and miR-150 rescue investigations. However, the authors should have put more focus on the general disease and/or biological relevance.*

Our response: We thank the referee for acknowledging our experimental effort and we apologize for the confusion regarding the objectives of this study.

Three years ago, we detected a decrease in the fraction of non-classical monocytes in the peripheral blood of CMML patients. This phenotype can be used routinely to distinguish a reactive monocytosis from a CMML (editorial by Peter L Greenberg in *Blood* in 2017 “The classical nature of distinctive CMML monocytes. *Blood* 129, 1745–1746 (2017)”). We noticed that a clinical and biological response to demethylating drugs was associated with the restoration of a normal monocyte subset repartition (Selimoglu-Buet D et al, *Blood* 2015), even though the mutation allele burden in monocytes remained unchanged (Merlevede J et al, *Nature Commun* 2016).

We started the present study with a blind screen of miRNA expression in patient monocytes, identifying miR-150 as the most deregulated miRNA. We were initially puzzled by these results as miR150, which had been involved in B and NK cell development and in erythroid versus megakaryocyte differentiation, had no identified function in the granulomonocyte lineage. As ageing is an important component of CMML pathophysiology (Merlevede J et al, *Nature Commun* 2016), we kept *Mir150*^{-/-} mice ageing for up to 2 years and never observed a CMML-like phenotype in these animals. In other words, miR-150 deletion by itself does not generate a CMML-like disease.

During these experiments, we noticed an abnormal repartition of monocyte subsets in *Mir150*^{-/-} mice. We took advantage of these results to explore the role of miR-150 in the generation of minor monocyte subsets. As indicated in the title, our manuscript is focused on the role of miR-150 in the generation of mouse and human non-classical monocyte subset, CMML being used as a model disease to explore the regulation / deregulation of miR-150 expression. We made some changes in the manuscript to clarify this point.

Modifications:

We modified the highlights to focus on the role of miR-150 in monocyte minor subset generation

- miR150 contributes to the conversion of classical into nonclassical monocyte subset

- Ten-eleven-translocation-3 (TET3) is a miR-150 target involved in the generation of nonclassical monocyte subsets.

- A lineage specific regulatory region is hypermethylated in classical monocytes of chronic myelomonocytic leukemia patients

In the discussion, we state that “The abnormal repartition of monocyte subsets that characterizes CMML has provided a diagnostic tool in patients with a monocytosis to rapidly distinguish this myeloid malignancy from a reactive monocytosis. This characteristic feature also provides new insights into circulating monocyte biology by identifying the role of a *miR-150/TET3* axis in the development of nonclassical monocyte subsets.”

To enforce our message, we have now performed transfer experiment whose results are shown on **figure 3, panels I and J** and described at the end the “*mmu*-miR-150 has a cell-autonomous effect on mouse monocyte subset generation” chapter:

Modifications: “Since the phenotype of *Mir150*^{-/-} mice is not the complete reduction of non-classical monocytes observed in *Nr4a1*^{-/-12,13} or *Cebpb*^{-/-36,37} mice, we explored the conversion ability of Ly6C^{high} by transfer experiment. A 1 to 1 ratio of wild-type CD45.1 and *Mir150*^{-/-} CD45.2 Ly6C^{high} monocytes were injected into transgenic mice that express Green Fluorescent Protein (GFP) under the direction of the human ubiquitin C promoter.³⁸ Forty hours later, we analyzed monocyte subsets by flow cytometry in the peripheral blood, the spleen and the bone marrow. Dead cells and cells expressing Ly6G, CD3, B220 and NK1.1 were excluded. Ly6C^{high} and Ly6C^{low} subsets were quantified in CD45.1⁺ and CD45.2⁺, GFP⁻, CD115⁺, CD11b⁺ monocyte populations. A defective conversion of Ly6C^{high} into Ly6C^{low} subset was observed in every studied tissue among *Mir150*^{-/-} CD45.2 compared to wild-type CD45.1 monocytes (**Figure 3I, 3J, and S5**). Of note, the 1/1 ratio of wild-type and *Mir150*^{-/-} monocytes at injection was conserved at 40 hours, indicating that *Mir150* gene deletion did not promote apoptosis in this time frame.”

Comment from the reviewer: (2) For instance, does miR-150 overexpression and the resulting formation of more non-classical monocytes have any consequence for the disease biology of CMML?

Our response: We have performed miR-150 overexpression experiments in mice. We had not included these results in the revised manuscript as, when overexpressed in mouse lin⁻ cells before engraftment, miR-150 provokes a severe anemia, which may be a consequence of miR-150 function in the fate of erythroid and megakaryocyte progenitors, as described in Lu J et al, Dev Cell. 2008;14:843 and in Sun Z, et al Oncotarget. 2015;6:43033. Results are further discussed and shown below in response to another comment by this reviewer.

We also overexpressed miR-150 in human CD34⁺ cells *ex vivo* and cultured these cells in liquid medium to generate monocytes, showing an increased production of CD14⁺,CD16⁺ cells (**figure 4C, 4D and S6C, S6D**). These experimental conditions did not generate enough cells to explore their biological properties in more details, but RNA sequencing and gene ontology analyses described above provide new information that guide ongoing investigation of monocyte biology in CMML.

Comment from the reviewer: *Specific major pitfalls: (1) How can the authors explain the huge variances of the miRNA expression levels between their Learning and Validation cohort (especially for miR-451...)?*

Our response: This is a very important comment, pointing to an ambiguity in the initial version of figure 1. Regarding miR-451, we have checked all our data and repeated some qRT-PCR measurements in an extended validation cohort. The main contradiction was in healthy donor samples as, with the internal control used in Figure 1 (*RUN6B*), the mean level of *has-mir451* expression appeared to be lower in healthy donor samples of the validation cohort than in those of the learning cohort. This discrepancy had not been detected with another internal control (*HPRT*). These ambiguous results could have been related to an insufficient number of controls in the two cohorts. The learning cohort could not be extended but we extended the validation control from 9 to 24 healthy donor samples and 42 to 53 CMML samples.

Modifications. The new version of **Figure 1 and S1** confirms that there is no significant change in miR451 expression in patient compared to healthy donor samples.

Comment from the reviewer: *And does the presented data fit to a recent publication by Zawada et al. (Immunobiology 2018)? On page 18, the authors state that miR-150 has been shown to be lower expressed in classical than in non-classical monocytes. They cite Zawada et al. But within this publication, non-classical miR-150 expression is higher than in intermediate monocytes and similar to that of classical monocytes. Regarding the huge variation within the authors initial data from Figure 1, the conclusion are somehow difficult. This should have been at least discussed.*

The comprehensive miRNA profiling performed by Gunnar Heine's group using small RNA sequencing (Zawada et al., Immunobiology 2018) detected multiple changes in miRNA expression among healthy human monocyte subsets, including an important decrease in miR-150-5p expression in intermediate compared to classical and non-classical monocytes. Discrepancies with our results may be due to the method used (RNA sequencing versus real-time quantitative PCR with 3 independent housekeeping genes). The method we used for RNA Sequencing did not allow adequate analysis of miRNA expression, precluding any comparison. Discrepancies could also be related to the heterogeneity of intermediate monocytes, as suggested by recent studies at the single cell level (*e.g.* Villani AC et al. Science. 2017;356:6335), indicating that differential gating and cell sorting strategies could have important consequences on miR-150 expression measurement. During the course of the revision of this manuscript, we performed new analyses in intermediate and nonclassical monocytes of healthy donor and CMML patients, which confirmed our previous results.

Modifications: Following the reviewer recommendation, we have better discussed these discrepancies in the new version of the manuscript :” Among circulating monocytes, the

expression of miR-150 was observed to be lower in classical than in nonclassical monocytes,^{49,50} with some discrepancies regarding intermediate monocytes⁵¹ that could be related to the heterogeneity of this subset.⁵²

Comment from the reviewer: (2) In Figure 3H: It is unclear why the authors just compared GFP+ to GFP- cells. The transduction itself can have a huge impact on cell behavior. A scrambled miRNA or GFP only coding vector should have been used (like for the human system in figure 4)

Our response: This important point also needed clarification.

As discussed above, we initially transduced lin⁻ bone marrow cells from *miR-150*^{-/-} mice with a retroviral vector encoding miR-150 and Green-Fluorescent Protein (GFP) or GFP alone. GFP-expressing cells were sorted, checked for miR-150 expression and transplanted to lethally irradiated WT recipients. In two independent preliminary experiments, all the mice transplanted with GFP-only lin⁻ cells survived, but 11 out of 13 transplanted with lin⁻ cells overexpressing miR-150 developed a severe anemia and died a few days post-transplantation, probably as a consequence of miR-150 function in erythromegakaryocytic commitment (Lu J et al, Dev Cell. 2008;14:843; Sun Z, et al Oncotarget. 2015;6:43033). Two animals survived six weeks post-transplantation, which allowed detecting an increase in their peripheral blood platelet count and a decrease in their Ly6C^{high}/ Ly6C^{low} monocyte subset ratio (compared to mice transplanted with GFP⁺ miR-150^{-/-} lin⁻ cells), further supporting the contribution of miR-150 to the conversion of Ly6C^{high} into Ly6C^{low} monocytes. Because of the interference with erythroid cell production, we did not report these preliminary results in the manuscript

Monocyte fractions are shown for the reviewer.

A retroviral vector encoding GFP alone or GFP and miR-150 was transduced in Mir150^{-/-} bone marrow lin⁻ cells. These cells were sorted on GFP expression and engrafted in mice. Monocyte subset repartition was examined 6 weeks post-engraftment. GFP-only group, n=6; miR-150 rescue group, n=2 (all the other animals died rapidly, see above).

Comment from the reviewer: (3) In Figure 4: The results from Figure 4C do not match with the bar graphs in Figure 4D. The FACS plots show 80% CD14⁺/CD16⁻ cells after miR-150 overexpression (the highest percentage of all conditions). In contrast, they show 20% CD16⁺/CD14⁺ in the miR-150 overexpression condition. This is the lowest percentage of all conditions. Within the bar graphs, it is exactly the opposite. How can this be explained?

Our response: Thank you for having detected this mistake, we apologize for having inverted the plots in the submitted version of **Figure 4C**.

Modifications: This error has been corrected in the revised version.

Comment from the reviewer: (4) In Figure 5A. Responders show accelerated miR-150 expression in CD14⁺ cells. How were the FACS panels gated? Does it include all CD14^{low} nc-monocytes? As the authors showed an increased expression of miR-150 in nc-monocytes compared to normal monocytes, these results are not surprising and might be independent of the demethylative activity of the DAC treatment. Moreover, it might just verify the results of Figure 4b. If you have more nc monocytes, miR-150 expression is higher.

Our response: It is an interesting suggestion that the increased expression of mir-150 could indicate an increase in non-classical monocytes but, as suggested by the reviewer, miR-150 expression was measured by qRT-PCR in sorted CD14⁺ cells, which eliminates a majority of CD14^{low} non-classical monocytes. We also provide in the following panels the demonstration that methylation of a myeloid lineage specific regulatory element called R3 region, which may be responsible for the decreased expression of the gene in patient samples, is partially reversed in responding patients, not in those who resist to treatment.

We also sorted monocyte subsets from the peripheral blood of 8 healthy donors and 10 CMML patients and performed qRT-PCR analysis of miR150 and 3 housekeeping gene expression. We confirmed the decreased expression of miR150 in CMML patient classical monocytes and found a similar result in intermediate monocytes. Importantly, even in CMML patient non-classical monocytes, we observed a higher expression of miR150 as compared to classical and intermediate subsets (new **figure 7B**). This could indicate that a subpopulation of leukemic monocytes has conserved a normal miR150 expression and regulation. Exploring this hypothesis will require single cell epigenetic analyses.

Modifications: Interestingly, while the expression of miR-150 was decreased in sorted classical and intermediate monocyte subsets collected from 10 CMML patients before any treatment and compared to healthy donor monocyte subsets, its expression was not decreased in the rare, residual nonclassical monocytes in CMML patients (**Figure 7D, S9B & S9C**), suggesting that a fraction of leukemic cells may escape the epigenetic down-regulation of *MIR150* gene in these patients.

Comment from the reviewer: (4) It would have been more interesting, if endogenous miR-150 expression in CD14⁺ cells before DAC treatment could predict responsiveness.

Our response: The global DNA methylation pattern can be predictive of CMML phenotype (*e.g.* Palomo L et al, Epigenetics. 2018;13:8). So far, we did not detect a predictive value of individual gene methylation status. In a previous study, we identified *TRIM33* gene promoter methylation in 35% of CMML samples but the expression of this gene was not predictive of response. Again, the restoration of the gene expression was associated with the response to hypomethylating drugs (Aucagne R et al, J Clin Invest 2011). We were also involved in the study published by Meldi K et al (J Clin Invest. 2015;125:1857) identifying differentially methylated regions (DMRs) of DNA at baseline that distinguish responders from nonresponders to hypomethylating drugs. These DMRs were primarily localized to nonpromoter regions, overlapping with distal regulatory enhancers, and did not include *MIR150* gene region. The 33 patient series that we have is too small to explore the hypothesis that *MIR150* promoter methylation could predict outcome: the question will be addressed in an ongoing European prospective clinical trial (ClinicalTrials.gov Identifier: NCT02214407).

Comment from the reviewer: (5) Figure 6: How is the methylation status of the R3-MIR150 promoter in nc monocytes compared to normal monocytes (either in healthy cells or in CMML, taken into account that nc monocytes are decreased in CMML)?

Our response: We provide a **new Figure 7A** showing that R3 promoter methylation is similar in the three monocyte subsets, which indicates that epigenetic deregulation of *MIR150* gene expression is observed only in a disease setting, in accordance with DNA aberrant methylation commonly observed in CMML, whereas physiological regulation may involve other mechanisms.

Modifications: Results are shown as a new **figure 7A**. “We also compared R3 methylation in sorted classical, intermediate and nonclassical healthy donor monocytes, which showed that the differential expression of miR-150 among these subsets did not depend on R3 methylation (Figure 7A).”

Comment from the reviewer: (6) A luciferase assay with mutations within the specific binding site(s) is necessary to prove the direct interaction of miR-150 and TET3 3'UTR in monocytic cells (e.g. U937).

Our response: The ability of miR-150 to regulate *TET3* expression through binding with its 3'-untranslated region (3'-UTR) had been demonstrated in 293T cells. The 3'-UTR of *TET3* predicted to interact with miR-150 had been cloned into a pMIR reporter luciferase vector and co-transfected with synthetic miR-150 into 293T cells. A marked reduction in the luciferase activity had been observed when *TET3* construct was co-transfected with synthetic miR-150, but not with the scrambled oligonucleotides. Random mutations in the recognition sequence suggested by Targetscan in 3'-UTR (RNA hybrid software suggests more than 10 other potential sites in this region) resulted in impairment of the reporter inhibition by miR-150 and the observed luciferase activity reduction was abrogated when the two sequences suggested by Targetscan as miR-150 interaction sites in 3'-UTR of these targets were mutated. These results have been published by Fang, Z. H. *et al.* (miR-150 exerts antileukemic activity in vitro and in vivo through regulating genes in multiple pathways. *Cell Death Dis.* 2016, 7, e2371) and the figure is shown below.

To answer the request of the reviewer, we have performed experiments in U937 human monocytic cells. WT and MUT target sequences (mutation of the two TET3 predicted targets indicated by Targetscan as above) were cloned in a GFP reporter plasmid using XmaI and Sall restriction sites. U937 cells (300,000 cells in 300 µl culture medium) were transduced with two lentiviruses: One encoded a GFP reporter and either WT or mutated TET3 target sequences (MOI 5) and the other one expressing or not miR-150 (MOI 20). GFP expression was assessed by flow cytometry 48 hours later. Though the experimental conditions were not those tested by Zhang et

al (smaller construct, no synthetic miR-150), the decrease in GFP signal induced by coexpression of miR150 and wildtype *TET3* reporter (between 27 and 30%) was less important when the two putative target sequences were mutated (between 12 and 21%). Since the demonstration has already been published, we suggest not providing these results in this manuscript.

Our response to Reviewer #3

Comment from the reviewer: *The manuscript by Selimoglu-Buet et al, proposes a regulatory role for miR-150 in generation of non-classical monocytes via targeting TET3 mRNA with a focus on CMML. The main concept of the manuscript is to assess the role of miR-150 in the proportional change of classical and non-classical monocytes in CMML patients. The study does not contain a lot of novelty compared to the publication of the same group in the manuscript of 2016 entitled "MIR150 Is Involved in the Monocyte Subset Differentiation and Its Down-Regulation Leads to Classical Monocyte Accumulation in Chronic Myelomonocytic Leukemia". The manuscript is a follow up study with additive experiments.*

Our response: The mentioned publication (Blood 2016 128:1133) is the abstract of a communication at the American Society of Hematology meeting in 2016. It is indicated on Nature Communications website that "abstracts do not compromise novelty". The data provided in this manuscript have never been published.

Comment from the reviewer: *In the first section of the results, the authors report on the down-regulation of miR-150 expression in peripheral blood monocytes from CMML patients as compared to healthy individuals. The comparison was carried out on CD14⁺ monocyte only, however the down-regulation of miR-150 may result from the different proportion of the monocyte sub-types (CD14⁺/CD16⁺) rather than expression of miR-150 itself. It is necessary that the authors analyze the expression of miR-150 in different subtypes: intermediate and the non-classical monocytes between CMML and healthy subjects.*

Our response: We agree with the reviewer and we performed the requested experiment. We sorted monocyte subsets from the peripheral blood of 8 healthy donors and 10 CMML patients and performed qRT-PCR analysis of miR150 and 3 housekeeping gene expression. This experiment demonstrated the decreased expression of miR150 in CMML patient classical and intermediate monocytes. Importantly, in the rare CMML patient non-classical monocytes, we still observed an increase in the expression of miR150 as compared to classical and intermediate subsets. This could indicate that a subpopulation of leukemic monocytes (our previous studies have shown that virtually every monocyte belongs to the leukemic clone) has conserved a normal miR150 expression, which could indicate subclonal epigenetic heterogeneity. Exploring this hypothesis will require single cell analyses.

Modifications: Results of miR150 expression level in healthy donor and CMML monocyte subsets are shown as a new **figure 7D** and discussed in the manuscript. "Interestingly, while the expression of miR-150 was decreased in sorted classical and intermediate monocyte subsets collected from 10 CMML patients before any treatment and compared to healthy donor monocyte subsets, its expression was not decreased in the rare, residual nonclassical monocytes in CMML patients (**Figure 7D, S9B & S9C**) suggesting that a fraction of leukemic cells may escape the epigenetic down-regulation of *MIR150* gene in these patients."

Comment from the reviewer: *The main phenotype of miR-150 'de-regulation' is the proportional shift (repartitioning) between classical vs non-classical monocytes in CMML vs healthy. This proportional shift could simply be due to a significant influx of immature CD14⁺/CD16⁻ monocytes*

in the blood or alternatively, the CD16+ cells migration into tissues. Both scenarios would result in repartitioning of the sub-types but not necessarily the differentiation interruption.

Our response: The reviewer raises a very important and exciting question. A part of the answer is in the transfer experiments we have performed in response to a question raised by reviewer 1. We wanted to determine if *miR-150*^{-/-}. Ly6C^{hi} were partially blocked in their conversion abilities. Results are shown in the **new figure 3, panels I and J**. They are described at the end the “mmu-miR-150 has a cell-autonomous effect on mouse monocyte subset generation” chapter.

Modifications: “Since the phenotype of *Mir150*^{-/-} mice is not the complete reduction of non-classical monocytes otherwise observed in *Nr4a1*^{-/-12,13} or *Cebpb*^{-/-36,37} mice, we explored the conversion ability of Ly6C^{high} by transfer experiment. A 1 to 1 ratio of wild-type CD45.1 and *Mir150*^{-/-} CD45.2 Ly6C^{high} monocytes were injected into transgenic mice that express Green Fluorescent Protein (GFP) under the direction of the human ubiquitin C promoter.³⁸ Forty hours later, we analyzed monocyte subsets by flow cytometry in the peripheral blood, the spleen and the bone marrow. Dead cells and cells expressing Ly6G, CD3, B220 and NK1.1 were excluded. Ly6C^{high} and Ly6C^{low} subsets were quantified in CD45.1⁺ and CD45.2⁺, GFP⁻, CD115⁺, CD11b⁺ monocyte populations. A defective conversion of Ly6C^{high} into Ly6C^{low} subset was observed in every studied tissue among *Mir150*^{-/-} CD45.2 compared to wild-type CD45.1 monocytes (**Figure 3I, 3J, and S5**).”

Of note, the CD45.1/1 / CD45.2/2 ratio was conserved at 40 hours, indicating that the lack of miR150 expression did not promote CD45.2/2 cell apoptosis in this time frame, which was another potential explanation to the abnormal repartition of monocyte subsets.

Modifications: “Of note, the 1/1 ratio of wild-type and *Mir150*^{-/-} monocytes at injection was conserved at 40 hours, indicating that *Mir150* gene deletion did not promote apoptosis in this time frame.”

Comment from the reviewer: *What is the proportion of CD34+ cells in the blood of CMML patients compared to healthy individuals? The authors should elaborate on this?*

Our response: Regarding the proportion of CD34⁺ cells in the blood of CMML patients compared to healthy individuals, our unpublished data, which are in accordance with those obtained by other groups in the field, indicate that, on the contrary to myeloproliferative neoplasms such as chronic myeloid leukemia and primary myelofibrosis, they are in the same range, excepted in patients developing a severe myelofibrosis, which is a rare situation.

Comment from the reviewer: *In the knock-down experiment using shRNA against miR-150, specifically on fig4-C, how come that sh-MIR150 treated CD34+ cells resulted in higher CD16+ cells as compared to NT. The same applies to miR150 over-expression plot! Intuitively, one would expect the opposite? Can the authors explain this?*

Our response: We apologize for this mistake (we have inverted the plots in the submitted version of Figure 4C) and thank the reviewer.

Modifications : The error has been corrected in the revised version.

Comment from the reviewer: *In the ChIPseq experiments in fig5-C, the authors showed that different histone marks are having different coverage across different cell types including CD14+ monocytes, T cells, NK cells and B cells. The authors did not provide the same data for the CD16+ monocytes as one of the main cells being under investigation. The authors should perform chip-seq data for CD16+/CD14- cells? This would show the epigenetic makeup of the poised R3 region in CD16+ cells.*

Our response: We have now performed ChIP-Seq experiments with an anti-H3K4me3 antibody in sorted classical, intermediate and non-classical monocyte subsets collected from 2 healthy donors. These data are included as **Figure 7B**. We also collected results from previously reported ChIP-Seq experiments with anti-H3K27Ac and H3K4me1 antibodies in classical and nonclassical monocytes (Schmidl C, et al. Transcription and enhancer profiling in human monocyte subsets. Blood. 2014;123:e90-9) and added our own H3K4me3 ChIP-seq to generate **Figure 7C**. We also performed R3 bisulfite analysis in sorted classical, intermediate and non-classical monocyte subsets to show the epigenetic makeup of R3 region in CD16+ cells. These data are shown as **Figure 7A and Supplemental Figure S9A**.

Modifications: The requested information is shown in a new **Figure 7**. “We also compared R3 methylation in sorted classical, intermediate and nonclassical healthy donor monocytes, which showed that the differential expression of miR-150 among these subsets did not depend on R3 methylation (**Figure 7A**). ChIP-seq experiments did not detect any difference in H3K4me1, H3K27Ac and H3K4me3 marks at R3 among monocyte subsets (**Figure 7B, 7C and S9A**).⁴²”

Comment from the reviewer: *In the ChIPseq experiments in fig5-C-D, what are the normalized peak counts/numbers for each histone mark in healthy vs CMML CD14+ monocytes? The authors should provide numbers and statistical analysis. Are the histone marks differentially occupied in R3 region across replicates, and in CMML vs healthy individuals? The authors should show the epigenetic makeup of promoter of FCGRT gene if R3 is in fact miR-150 related promoter? The authors also need to provide the input tracks of each mark related to each sample to document the actual signal-to-noise ratio for different histone marks.*

Our response: We have normalized the results of ChIP-seq experiments for each histone mark and provide requested information. Figures have been corrected to show normalized Bigwig. Normalized peak counts/numbers for each histone mark are also provided. **Supplemental S7B** shows H3K4me3 at the R3 level in healthy donor and CMML patient monocytes, showing an equal enrichment, without any statistical difference. The input track of each sample is now included. Because we did not see any impact of CRISPR/Cas9 and R3 methylation in CMML samples on FCGRT gene expression, we did not analyze the epigenetic makeup of FCGRT promoter

Modifications: The requested information is shown in **Figure 5C, 5D** and **supplemental Figure S7**. “ChIP-seq experiments confirmed that H3K4me3 mark was located on R3 in healthy donor monocytes as well as in those collected from CMML patients (**Figure 5D**; peak calling with a p-value of 0.05: 107,503, 119,009 and 144,139 in control samples, 64,730, 79,133, 134,677, 290,419 in CMML samples) with an equal enrichment in control and CMML samples (**Figure S7B**). “

Comment from the reviewer: In the knock-out experiment fig5-E-F, how do the authors explain that miR-150 is expressed after deletion of R3 as it is suggested that it contains the miR-150 promoter? The presence of H3K4me1 and H3K27ac over R3 suggests that it could very well be the active enhancer rather than a promoter in monocytes. The authors should proof with additional experiments that R3 is the promoter.

Our response: The reviewer raises to several important points.

First, CRISPR/Cas9 deletion of R3 region was only partial, which explains why a signal could be detected by qRT-PCR after deletion. To clarify this point, we have included a snapshot of the R3 region indicating the deleted zone as **Figure S7D**.

Modifications: We now indicate that CRISPR/Cas9 deletion of R3 region was only partial, which explains why a signal can still be detected by qRT-PCR with primers.

“To validate that R3 could be an active *MIR150* promoter, we used CRISPR-Cas9 technology to delete a part of R3 region in U937 and K562 myeloid cell lines (**Figure S7C, S7D**). Compared to wildtype clones, clones in which R3 has been partially deleted showed a down-regulation of hsa-miR-150 expression without any change in *FCGRT* gene expression whose transcriptional start site is close to R3 sequence, further supporting a role for R3 as a *MIR150* specific promoter in these cells (**Figure 5E, 5F, S7E, S7F and S7G**).”

Regarding the promoter nature of R3, the presence of H3K4me3 combined with H3K27Ac usually marks accessible promoters with ongoing transcription (Saeed S et al. Science 2014). Also, R3 is a CpG rich region, like the majority of human gene promoters (Lenhard B et al. Nat Rev Genet 2012), whereas enhancers, at least transcribed ones, are usually CpG poor (Andersson R et al. Nature 2014).

miRNA encoding genes are difficult to analyze as they are transcribed into pri-miRNAs whose rapid processing by Drosha precludes easy identification with conventional sequencing techniques. A recent report indicates that one third of pri-miRNA have multiple potential TSS, in both inter- and intragenic loci, most of them at a 10^3 – 10^5 bp distance of the miRNA gene (Bouvy-Liivrand, M., et al. Nucleic Acids Research, 2017). GRO-cap has been used to capture TSS information from nascent transcripts in K562 myeloid cells and in GM12878 B cell line. Using GSE60456 accession, we have loaded these data and examined *MIR150* and *FCGRT* loci. We observed that the active TSS in K562 myeloid cells was distinct from that detected in the lymphoblastoid cell line, as illustrated below (orange frames).

GRO-cap analysis of TSSs in a human myeloid (K562) and a human lymphoid (GM12878) cell line. IGV shot of MIR150 / FCGRT locus. Red, plus strand; blue, minus strand. The bed files of all TSS found in K562 and GM12878 are shown. R3 region is shown.

GRO-Seq data generated in another myeloid cell line, THP-1, show some reads around R3. The coverage is not sufficient but results suggest the transcription of full-length pri-miR150 could start at R3.

Modifications: The requested information are shown in a new supplemental **Figure S7A**. “H3K27ac and H3K4me3 mark location suggested that promoter region 1 (R1) was active in T, B and NK cells but not in monocytes in which these marks overlapped in a distinct, CpG-enriched region called region 3 (R3). CpG enrichment combined with H3K4me3 and H3K27Ac marks suggested that R3 was a promoter with ongoing transcription.^{40,41} GRO-cap analyses (GSE60456 accession) identified a distinct TSS on the minus DNA strand in K562 myeloid cells and GM12878 B-cells, the TSS being located on R3 in K562 cells (**Figure S7A**).”

Comment from the reviewer: 10- To strengthen the suggested direct interaction of miR-150/TET3 pathway in monocyte repartitioning, it the authors should perform a Tet3 rescue experiment in Tet3^{-/-} cells to see if that can change the proportional phenotype back to normal as was done for miR-150 rescue experiment.

Our response: We get access to Tet3^{-/-} animals by collaborating with Olivier Bernard’s group in Gustave Roussy. Extensive breeding of these animals appears to be challenging, and the group is currently analyzing the phenotype of these animals to submit a manuscript soon. Therefore, the proposed rescue could not be performed in the time frame of this review.

Comment from the reviewer: Minor point, page 16, line 8-11 “ We cultured these cells in liquid medium with stem cell factor and M-CSF for 5 to 9 days, before analyzing, after having excluded CD71⁺ or CD163⁺ macrophages, the percentage of CD14⁺CD16⁻ and CD14⁺CD16⁺ Generated monocytes ” is unclear and needs to be rephrased.

Modifications: We have rephrased the sentence. The gating strategy is described in the legend of the figures and the sentence is now: “We cultured these cells in liquid medium with stem cell factor and M-CSF for 5 to 9 days before measuring the fraction of cells with a CD14⁺CD16⁻ and a CD14⁺CD16⁺ phenotype (**Figure 4C and S6C, S6D**)”

Comment from the reviewer: *Minor point, the figures are in general not immediately informative. For example, changing the color of H3K27ac track in figure 5-C and correcting the upside-down x axis labels in figure S3-S7.*

Modifications: We have performed the suggested change in the color of H3K27ac in figure 5C and thank the reviewer for these useful advices.

Figure S3 and S7 were in a landscape configuration, which explains why the x axis labels look upside-down. We have changed the orientation to put them in portrait mode.

REVIEWERS' COMMENTS:

Reviewer #1 (Remarks to the Author):

The authors provide a substantial amount of new data and sufficiently answered all my questions. I hereby recommend publication.

Only a few notes:

1. Can the authors provide the data for the micro-RNA pull down experiment in a supplementary table?
2. Typo page 15, line 334: 'miR150-/- gene deletion did not significantly change the expression of Cd115, Cebpa and Nr4a1' Shouldn't this be Cebpb?
3. Typo page 30, line 634: 'has-miR150'
4. Typo page 30, line 649: 'four paire sticky end'
5. Concerning the figure legends: You abbreviate 'wildtype (WT)' three times in figure legend 2.
6. For figure 9: Can you depict in the vulcano blot at least the 10 most significantly up- and down-regulated genes and can you comment on the fact that Tet3 is not one of them.
7. Check spelling of Cebp β (correct for protein: C/EBP β)

Reviewer #1 comments on revisions made in response to the original Reviewer #2 (Remarks to Author):

Regarding the first comment:

"(1) Most of the presented data are mainly descriptive. What is the rationale in miR-150 depletion for the function of non-classical monocytes within CMML or healthy cells? Is there any biological consequence? The miR-150 KO effects are quite mild and do not completely prevent nc monocyte formation. The reduction from 20% to 10% of CD14+ cells illustrates that miR-150 is not essential for nc monocyte formation, but reduces the total amount of cells. Are the remaining 10% nc monocytes (without any miR-150) biologically different from miR-150 wildtype nc monocytes?" There is a discrepancy in the understanding about the motivation/intention of the paper. Reviewer 2 is interested in any clinical relevance. For instance: 'Does the reduced number of non-classical monocytes lead to a better or worse outcome for the patient? What do these monocytes do during the disease? Etc...' Answers to these questions are almost impossible to provide, especially if there is no animal model for CMML. And as the authors state in their response to this comment:

"We questioned the mechanisms that control monocyte subset generation, taking advantage of the abnormal repartition of monocyte subsets in CMML." and also later:

"As indicated in the title, our manuscript is focused on the role of miR-150 in the generation of mouse and human non-classical monocyte subset, CMML being used as a model disease to explore the regulation / deregulation of miR-150 expression."

Therefore the authors are not directly interested in the function of non-classical monocytes during CMML, they are interested in the conversion process of classical to non-classical monocytes, which can be used for diagnostics. And for this they provide a huge amount of non-descriptive data.

The second comment follows the same line:

"(2) The reviewer is really honoring the huge experimental effort within the in vivo animal model and the time-consuming retransplantation and miR-150 rescue investigations. However, the authors should have put more focus on the general disease and/or biological relevance. For instance, does miR-150 overexpression and the resulting formation of more non-classical monocytes have any consequence for the disease biology of CMML?"

Again, in my perspective, these questions are relevant, but impossible to tackle, especially without a good CMML mouse system. The reviewer demands a non-classical monocyte-specific overexpression of miR-150 in vivo. To the best of my knowledge, I never saw a non-classical monocyte-specific overexpression of any gene. The authors tried to execute an experiment to answer his question and over-expressed miR150 in all hematopoietic cells, but due to miR150 function in other cells and the resulting anemia, they did not succeed:

"We have performed miR-150 overexpression experiments in mice. We had not included these results in the revised manuscript as, when overexpressed in mouse lin⁻ cells before engraftment, miR-150 provokes a severe anemia, which may be a consequence of miR-150 function in the fate of erythroid and megakaryocyte progenitors, as described in Lu J et al, Dev Cell. 2008;14:843 and in Sun Z, et al Oncotarget. 2015;6:43033. Results are further discussed and shown below in response to another comment by this reviewer"

Following comments:

"Specific major pitfalls: (1) How can the authors explain the huge variances of the miRNA expression levels between their Learning and Validation cohort (especially for miR-451...)?" and "And does the presented data fit to a recent publication by Zawada et al. (Immunobiology 2018)? On page 18, the authors state that miR-150 has been shown to be lower expressed in classical than in non-classical monocytes. They cite Zawada et al. But within this publication, non-classical miR-150 expression is higher than in intermediate monocytes and similar to that of classical monocytes. Regarding the huge variation within the authors initial data from Figure 1, the conclusion are somehow difficult. This should have been at least discussed."

These comments are important and some concerns between the two results obtained in different labs remain. However, the authors tried their best to answer this and performed new experiments that support their own results.

"In Figure 3H: It is unclear why the authors just compared GFP⁺ to GFP⁻ cells. The transduction itself can have a huge impact on cell behavior. A scrambled miRNA or GFP only coding vector should have been used (like for the human system in figure 4)"

Retroviral overexpression experiments are very difficult experiments and as stated correctly by the reviewer, can have a impact on cell behavior/differentiation. Furthermore, as previously mentioned, the experiment failed due to the possible function of miR150 in other hematopoietic cells. The authors provide the data for the reviewer.

"(4) In Figure 5A. Responders show accelerated miR-150 expression in CD14⁺ cells. How were the FACS panels gated? Does it include all CD14 low nc-monocytes? As the authors showed an increased expression of miR-150 in nc-monocytes compared to normal monocytes, these results are not surprising and might be independent of the demethylative activity of the DAC treatment. Moreover, it might just verify the results of Figure 4b. If you have more nc monocytes, miR-150 expression is higher."and "(4) It would have been more interesting, if endogenous miR-150 expression in CD14⁺ cells before DAC treatment could predict responsiveness."

This is a very important comment by the reviewer and an alternative explanation of the results. However, the authors performed a new experiment (new Figure 7B) in which they show that miR150 expression is decreased in classical monocytes from CMML. After DAC treatment, the expression is normalized and seen as an increase as depicted in Fig. 5A. Therefore I think the explanation provided by the authors is more consistent.

"We also sorted monocyte subsets from the peripheral blood of 8 healthy donors and 10 CMML patients and performed qRT-PCR analysis of miR150 and 3 housekeeping gene expression. We confirmed the decreased expression of miR150 in CMML patient classical monocytes and found a similar result in intermediate monocytes. Importantly, even in CMML patient non-classical monocytes, we observed a higher expression of miR150 as compared to classical and intermediate subsets (new figure 7B). This could indicate that a subpopulation of leukemic monocytes has conserved a normal miR150 expression and regulation. Exploring this hypothesis will require single cell epigenetic analyses."

"(3) In Figure 4: The results from Figure 4C do not match with the bar graphs in Figure 4D. The FACS plots show 80% CD14⁺/CD16⁻ cells after miR-150 overexpression (the highest percentage of all conditions). In contrast, they show 20% CD16⁺/CD14⁺ in the miR-150 overexpression condition. This is the lowest percentage of all conditions. Within the bar graphs, it is exactly the opposite. How can this be explained?"

Mistake by the authors and solved: "Thank you for having detected this mistake, we apologize for

having inverted the plots in the submitted version of Figure 4C."

"(5) Figure 6: How is the methylation status of the R3-MIR150 promoter in nc monocytes compared to normal monocytes (either in healthy cells or in CMML, taken into account that nc monocytes are decreased in CMML)?"

Sufficiently answered by the authors and now shown in the paper (new Fig. 7A).

"(6) A luciferase assay with mutations within the specific binding site(s) is necessary to prove the direct interaction of miR-150 and TET3 3'UTR in monocytic cells (e.g. U937)."

The authors refer to another paper, where they performed a similar and detailed luciferase assay for Tet3. They further tested this as suggested in U937 cells and present the results for the eye of the reviewer only. For completeness, the authors could have put more energy to answer this question. However, due to the existing data, the experiment shown for the reviewer is sufficient. Maybe this data set can be shown in the supplement.

Reviewer #3 (Remarks to the Author):

The authors have satisfactorily addressed the comments of the reviewers including mine. The only point that could not be addressed is the expression of exogenous TET in TET^{-/-} cells to assess whether the effects attributed to TET are indeed correct. The authors provided an acceptable explanation why this experiment could not be done at this point.

I believe that although this is an important point, the revised manuscript can be accepted.

For the reviewers,

Our point-by-point answers to your requirements are listed in blue below the reviewer's comments and corresponding changes have been made in the text.

Remarks by reviewer #1:

The authors provide a substantial amount of new data and sufficiently answered all my questions. I hereby recommend publication.

Only a few notes:

1. Can the authors provide the data for the micro-RNA pull down experiment in a supplementary table?

As proposed by the reviewer, we added the RNA identified by micro-RNA pull-down as a new Supplementary Table (Supplementary Table 2).

2. Typo page 15, line 334: 'miR150^{-/-} gene deletion did not significantly change the expression of Cd115, Cebpa and Nr4a1' Shouldn't this be Cebpb?

Indeed, we had made a mistake. This sentence has been changed.

3. Typo page 30, line 634: 'has-miR150'.

The error has been corrected

4. Typo page 30, line 649: 'four paire sticky end'.

The error has been corrected

5. Concerning the figure legends: You abbreviate 'wildtype (WT)' three times in figure legend 2.

We suppressed 2 of the 3 abbreviations.

6. For figure 9: Can you depict in the vulcano blot at least the 10 most significantly up- and down-regulated genes and can you comment on the fact that Tet3 is not one of them.

We have now annotated the most significantly modulated genes in figure 9 and changed the figure legend as follows: "Figure 9. RNA sequencing of mouse monocyte subsets Volcanoplot representation of differentially expressed genes in wild-type littermates versus either *Mir150^{-/-}* (upper panels) or *Tet3^{-/-}* (lower panels) monocytes subtypes. The number of differentially expressed genes (cut-off q-value 0.001) is indicated in each panel, as well as the name of differentially deregulated genes." We also added the following sentence in the text "Of note, *Mir150^{-/-}* gene deletion did not significantly change the expression of *Cd115*, *Cebpb*, *Nr4a1* and *Tet3* genes (**Supplementary Table 3**). This latter result could indicate that miR-150 targets Tet3 translation rather than *Tet3* gene transcription."

7. Check spelling of Cebpβ (correct for protein: C/EBPβ).

Done

Reviewer #1 comments on revisions made in response to the original Reviewer #2 (Remarks to Author):

Regarding the first comment: "(1) Most of the presented data are mainly descriptive. What is the rationale in miR-150 depletion for the function of non-classical monocytes within CMML or healthy cells? Is there any biological consequence? The miR-150 KO effects are quite mild and do not completely prevent nc monocyte formation. The reduction from 20% to 10% of CD14+ cells illustrates that miR-150 is not essential for nc monocyte formation, but reduces the total amount of cells. Are the remaining 10% nc monocytes (without any miR-150) biologically different from miR-150 wildtype nc monocytes?" There is a discrepancy in the understanding about the motivation/intention of the paper. Reviewer 2 is interested in any clinical relevance. For instance: 'Does the reduced number of non-classical monocytes lead to a better or worse outcome for the patient? What do these monocytes do during the disease? Etc...'

Answers to these questions are almost impossible to provide, especially if there is no animal model for CMML.

And as the authors state in their response to this comment:

- "We questioned the mechanisms that control monocyte subset generation, taking advantage of the abnormal repartition of monocyte subsets in CMML." and also later:
- "As indicated in the title, our manuscript is focused on the role of miR-150 in the generation of mouse and human non-classical monocyte subset, CMML being used as a model disease to explore the regulation / deregulation of miR-150 expression."

Therefore the authors are not directly interested in the function of non-classical monocytes during CMML, they are interested in the conversion process of classical to non-classical monocytes, which can be used for diagnostics. And for this they provide a huge amount of non-descriptive data.

We are grateful to the reviewer for agreeing with our response to initial comments by reviewer 2

The second comment follows the same line:

"(2) The reviewer is really honoring the huge experimental effort within the in vivo animal model and the time-consuming retransplantation and miR-150 rescue investigations. However, the authors should have put more focus on the general disease and/or biological relevance. For instance, does miR-150 overexpression and the resulting formation of more non-classical monocytes have any consequence for the disease biology of CMML?"

Again, in my perspective, these questions are relevant, but impossible to tackle, especially without a good CMML mouse system. The reviewer demands a non-classical monocyte-specific overexpression of miR-150 in vivo. To the best of my knowledge, I never saw a non-classical monocyte-specific overexpression of any gene. The authors tried to execute an experiment to answer his question and over-expressed miR150 in all hematopoietic cells, but due to miR150 function in other cells and the resulting anemia, they did not succeed:

"We have performed miR-150 overexpression experiments in mice. We had not included these results in the revised manuscript as, when overexpressed in mouse lin- cells before engraftment, miR-150 provokes a severe anemia, which may be a consequence of miR-150 function in the fate of erythroid and megakaryocyte progenitors, as described in Lu J et al, Dev Cell. 2008;14:843 and in Sun Z, et al Oncotarget. 2015;6:43033. Results are further discussed and shown below in response to another comment by this reviewer"

Again, we can only thank the reviewer for careful reading and analysis of the provided data and acknowledging the experimental limits to answer some of the questions raised.

Following comments:

"Specific major pitfalls: (1) How can the authors explain the huge variances of the miRNA expression levels between their Learning and Validation cohort (especially for miR-451...)"

and "And does the presented data fit to a recent publication by Zawada et al. (Immunobiology 2018)? On page 18, the authors state that miR-150 has been shown to be lower expressed in classical than in non-classical monocytes. They cite Zawada et al. But within this publication, non-classical miR-150 expression is higher than in intermediate monocytes and similar to that of classical monocytes. Regarding the huge variation within the authors initial data from Figure 1, the conclusion are somehow difficult. This should have been at least discussed." These comments are important and some concerns between the two results obtained in different labs remain. However, the authors tried their best to answer this and performed new experiments that support their own results.

"In Figure 3H: It is unclear why the authors just compared GFP+ to GFP- cells. The transduction itself can have a huge impact on cell behavior. A scrambled miRNA or GFP only coding vector should have been used (like for the human system in figure 4)" Retroviral overexpression experiments are very difficult experiments and as stated correctly by the reviewer, can have a impact on cell behavior/differentiation. Furthermore, as previously mentioned, the experiment failed due to the possible function of miR150 in other hematopoietic cells. The authors provide the data for the reviewer.

"(4) In Figure 5A. Responders show accelerated miR-150 expression in CD14+ cells. How were the FACS panels gated? Does it include all CD14 low nc-monocytes? As the authors showed an increased expression of miR-150 in nc-monocytes compared to normal monocytes, these results are not surprising and might be independent of the demethylative activity of the DAC treatment. Moreover, it might just verify the results of Figure 4b. If you have more nc monocytes, miR-150 expression is higher."and "(4) It would have been more interesting, if endogenous miR-150 expression in CD14+ cells before DAC treatment could predict responsiveness."

This is a very important comment by the reviewer and an alternative explanation of the results. However, the authors performed a new experiment (new Figure 7B) in which they show that miR150 expression is decreased in classical monocytes from CMML. After DAC treatment, the expression is normalized and seen as an increase as depicted in Fig. 5A. Therefore I think the explanation provided by the authors is more consistent.

"We also sorted monocyte subsets from the peripheral blood of 8 healthy donors and 10 CMML patients and performed qRT-PCR analysis of miR150 and 3 housekeeping gene expression. We confirmed the decreased expression of miR150 in CMML patient classical monocytes and found a similar result in intermediate monocytes. Importantly, even in CMML patient non-classical monocytes, we observed a higher expression of miR150 as compared to classical and intermediate subsets (new figure 7B). This could indicate that a subpopulation of leukemic monocytes has conserved a normal miR150 expression and regulation. Exploring this hypothesis will require single cell epigenetic analyses."

(3) In Figure 4: The results from Figure 4C do not match with the bar graphs in Figure 4D. The FACS plots show 80% CD14+/CD16- cells after miR-150 overexpression (the highest percentage of all conditions). In contrast, they show 20% CD16+/CD14+ in the miR-150 overexpression condition. This is the lowest percentage of all conditions. Within the bar graphs, it is exactly the opposite. How can this be explained?"

Mistake by the authors and solved: "Thank you for having detected this mistake, we apologize for having inverted the plots in the submitted version of Figure 4C."

"(5) Figure 6: How is the methylation status of the R3-MIR150 promoter in nc monocytes compared to normal monocytes (either in healthy cells or in CMML, taken into account that nc monocytes are decreased in CMML)?"

Sufficiently answered by the authors and now shown in the paper (new Fig. 7A).

Again, we thank the reviewer for the positive comments

"(6) A luciferase assay with mutations within the specific binding site(s) is necessary to prove the direct interaction of miR-150 and TET3 3'UTR in monocytic cells (e.g. U937)."

The authors refer to another paper, where they performed a similar and detailed luciferase assay for Tet3. They further tested this as suggested in U937 cells and present the results for the eye of the reviewer only. For completeness, the authors could have put more energy to answer this question. However, due to the existing data, the experiment shown for the reviewer is sufficient. Maybe this data set can be shown in the supplement.

As proposed by the reviewer, we show the results of this experiment in Supplementary Figure 10, panel A, and modified the text, supplementary methods, supplementary Figure 10A and its legend accordingly.

Reviewer #3 (Remarks to the Author):

The authors have satisfactorily addressed the comments of the reviewers including mine. The only point that could not be addressed is the expression of exogenous TET in TET^{-/-} cells to assess whether the effects attributed to TET are indeed correct. The authors provided an acceptable explanation why this experiment could not be done at this point.

I believe that although this is an important point, the revised manuscript can be accepted.